# OptiScene: LLM-driven Indoor Scene Layout Generation via Scaled Human-aligned Data Synthesis and Multi-Stage Preference Optimization

**Yixuan Yang**[1,3*]  **Zhen Luo**[2,1*]  **Tongsheng Ding**[1*]  **Junru Lu**[3,6]
**Mingqi Gao**[1]  **Jinyu Yang**[4]  **Victor Sanchez**[3]  **Feng Zheng**[1,5†]

[1]SUSTech    [2]Shanghai Innovation Institute    [3]University of Warwick
[4]Tapall.ai    [5]Spatialtemporal AI    [6]Tencent Youtu Lab

arnoldyang97@gmail.com, luoz2024@mail.sustech.edu.cn, a850538951@gmail.com

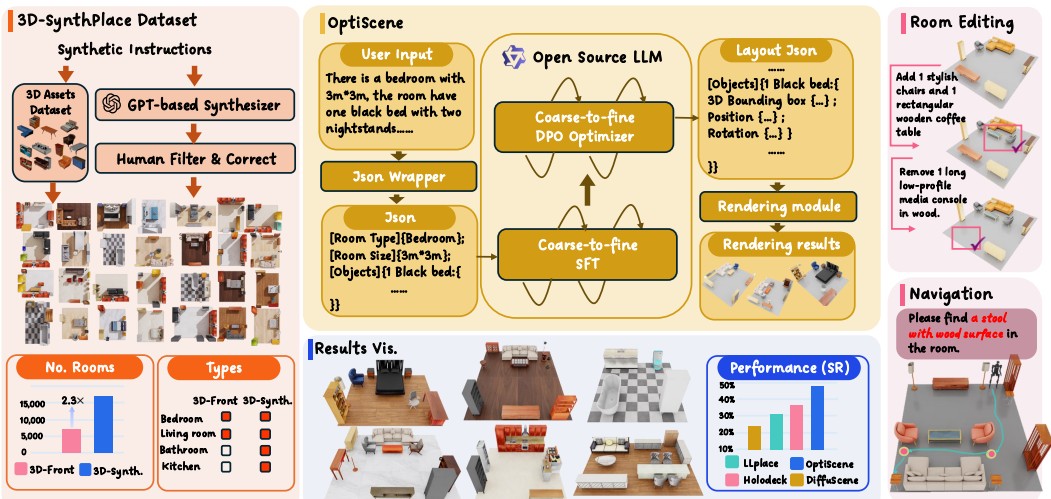

Figure 1: Overview of proposed **3D-SynthPlace** dataset and **OptiScene** framework for indoor layout generation. (Left): We propose 3D-SynthPlace, a large-scale, high-quality indoor layout dataset. (Middle): Our open-source LLM-based generator, OptiScene, takes user instructions and produces structured layout representations through a two-stage, coarse-to-fine optimization. (Right): OptiScene supports interactive layout editing and downstream tasks such as robotic navigation. (Bottom): Qualitative layout visualizations and quantitative comparisons show OptiScene's superior performance (sucess rate) over existing prompt-driven and learning-based baselines.

## Abstract

Automatic indoor layout generation has attracted increasing attention due to its potential in interior design, virtual environment construction, and embodied AI. Existing methods fall into two categories: *prompt-driven* approaches that leverage proprietary LLM services (e.g., GPT APIs), and *learning-based* methods trained on layout data upon diffusion-based models. Prompt-driven methods often suffer from spatial inconsistency and high computational costs, while learning-based methods are typically constrained by coarse relational graphs and limited datasets, restricting their generalization to diverse room categories. In this paper, we revisit LLM-based indoor layout generation and present **3D-SynthPlace**, a large-scale

---

[*]Equal contribution.
[†]Corresponding author.

39th Conference on Neural Information Processing Systems (NeurIPS 2025).

dataset that combines synthetic layouts generated via a 'GPT synthesize, Human inspect' pipeline, upgraded from the 3D-Front dataset. 3D-SynthPlace contains nearly 17,000 scenes, covering four common room types—bedroom, living room, kitchen, and bathroom—enriched with diverse objects and high-level spatial annotations. We further introduce **OptiScene**, a strong open-source LLM optimized for indoor layout generation, fine-tuned based on our 3D-SynthPlace dataset through our two-stage training. For the warm-up stage I, we adopt supervised fine-tuning (SFT), which is taught to first generate high-level spatial descriptions then conditionally predict concrete object placements. For the reinforcing stage II, to better align the generated layouts with human design preferences, we apply multi-turn direct preference optimization (DPO), which significantly improves layout quality and generation success rates. Extensive experiments demonstrate that OptiScene outperforms traditional prompt-driven and learning-based baselines. Moreover, OptiScene shows promising potential in interactive tasks such as scene editing and robot navigation, highlighting its applicability beyond static layout generation. Project page: *optiscene.github.io/*.

# 1 Introduction

Indoor Scene Layout Generation has attracted growing attention across the 3D vision fields, including virtual environment synthesis [40, 10, 46, 49, 4, 3, 12], robot navigation [29, 28, 50, 45], digital twin construction [5, 20, 30, 39], interior design automation [42, 14, 17], and human-centric simulation in some scenes [47, 43, 36, 21, 38, 16]. With the increasing need for simulating realistic virtual environments, learning to generate functionally plausible, physically feasible, and semantically meaningful spatial arrangements has become an essential capability for intelligent systems.

Existing approaches for indoor scene layout generation fall broadly into two paradigms: *prompt-driven methods* that leverage large language models (LLMs) through well-defined natural language prompts, and *learning-based methods* that train models on layout data to capture spatial priors. Prompt-driven methods, such as Holodeck [42], SceneCraft [14], FlairGPT [18], SceneTeller [23], I-Design [2], LayoutVLM [32] and LayoutGPT [8], typically rely on proprietary LLM services (e.g., GPT APIs) to produce layouts based on multi-turn prompting or retrieval-based in-context examples. Although prompting approaches have shown competitive performance, they suffer from fundamental limitations such as incorrect domain alignment, weak controllability, and most importantly, superficial physical understanding. Moreover, these commercial LLM APIs are closed-source, prohibitively expensive for large-scale generation, and thoroughly static that can not be modeled with new 3D spatial priors or perfectly encoded with human preferences. As a result, they are ill-suited for fine-grained layout tasks that require geometric validity, functional coherence, and task-specific adaptation. Learning-based methods, on the other hand, explicitly optimize local models using open-source structured layout data. Diffusion-based approaches such as LEGO-Net [37], DiffuScene [33], InstructScene [17], and PhyScene [40] refine layouts iteratively by modeling object-object relationships through layout graphs or bounding boxes. LLplace [41] represents another type of attempt to fine-tune LLMs instead of diffusion-based models. Most learning-based models are trained on datasets like 3D-Front [9], which contain fewer than 7,000 usable layouts and focus predominantly on bedrooms and living/dining rooms. This limited data diversity not only restricts scene coverage but also biases the model toward overfitting on narrow spatial patterns. Beyond the dataset, while existing learning-based methods have shown progress in modeling spatial layouts, they essentially lack *a preference-aware learning objective* that aligns with human minds. Most existing approaches optimize for data reconstruction or distribution matching, rather than aligning with human notions, such as what constitutes a desirable, functional, or aesthetically pleasing layout. Without explicit reasoning of human-aligned supervision, these models struggle to generalize across room types, adapt to variable orientations, or reflect higher-level spatial semantics—capabilities that are essential for practical deployment in design systems or embodied environments.

To address the fundamental obstacle—namely the lack of human-aligned optimization preference, insufficient spatial reasoning, and limited scene diversity—we propose **3D-SynthPlace**, a large-scale dataset with more than 16,000 layouts that augments 3D-Front [9] with over 9,000 synthetic layouts generated via Holodeck, as well as introduce **OptiScene**, an LLM-based framework that enables controllable and preference-aware indoor layout generation. Different from existing GPT-based pipelines that depend on closed APIs and prompt engineering, OptiScene is dedicated to post-tuning

open-source LLM for spatial design tasks, making it adaptable, efficient, and deployable in real-world systems. Serving as an important prerequisite, 3D-SynthPlace ensures sufficient training diversity. All generated scenes within the dataset are manually filtered and rigorously corrected to ensure quality, and the dataset expands room coverage beyond the conventional bedroom/living room focus to include kitchens and bathrooms, providing a more comprehensive training foundation. Building on 3D-SynthPlace, we enhance the regular SFT tuning by introducing high-level semantic reasoning plans: instead of directly predicting placements, the model is first guided to generate a natural language summary that describes spatial organization and object relationships inside the room. This step provides an interpretable outline that helps the model internalize functional zoning and spatial intent—challenges that graph-based or purely coordinate-driven approaches often fail to address. Afterwards, the SFT model utilizes the room layout summary for robust generation. To further align the generation with human preferences, we introduce a two-stage *Direct Preference Optimization (DPO)[26]* reinforcing process. In the first stage of DPO, we construct preference pairs from an expert-curated subset of human-consensus layouts, as the positive preference, and contrast them against a group of suboptimal model-generated variants by the SFT basis, fostering the SFT model to capture subtle stylistic and structural preferences. In the second stage, we synthetically inject spatial violations (e.g., collisions and boundary overflows) into positive scenes to produce harder negatives, encouraging the model to learn robust, geometry-aware layout decisions. Through this reasoning-guided, preference-aligned training strategy, our OptiScene bridges the gap between static language generation and dynamic spatial intelligence, producing layouts that are physically plausible, semantically coherent, and practically usable across design and robotics workflows.

As shown in Fig. 1, OptiScene demonstrates consistent improvements across multiple dimensions in our experiments, including success rate and layout quality. Overall, our contributions are threefold:

- We construct **3D-SynthPlace**, a large-scale dataset with more than 16,000 scene layouts, which combine the 3D-Front dataset and over 9,000 synthetic layouts generated by Holodeck. All samples are manually filtered and corrected to ensure quality, and we expand the scene coverage to include kitchen and bathroom layouts, providing a richer and more diverse training corpus for indoor layout generation.
- We propose **OptiScene**, a human-aligned indoor layout generation framework. The framework introduces high-level semantic reasoning into a supervised fine-tuning (SFT) stage, encouraging the model to infer spatial intent and object relationships before predicting concrete layouts. Building on this, it further applies a two-stage Direct Preference Optimization (DPO) reinforcement to align layout generation with human preferences and physical constraints.
- We conduct comprehensive evaluations across multiple metrics and prove that OptiScene achieves state-of-the-art performance. Furthermore, we show its effectiveness on downstream tasks like interactive editing and robot navigation, validating its practical applicability.

## 2   Related work

**Methods & datasets for scene layout generation.** Besides traditional methods [35, 34, 25], recent methods can be grouped into *prompt-driven* and *learning-based* approaches. Prompt-driven methods use LLMs via structured or free-form prompts to generate layouts, including Holodeck [42], SceneCraft [14], FlairGPT [18], SceneTeller [23], I-Design [2], Aguina [1], LayoutVLM [32], and LayoutGPT [8]. These methods show language-scene grounding but often rely on proprietary LLMs (e.g., GPT), lack spatial fine-tuning, and offer limited controllability or offline deployment. Diffusion models have been used in widely areas [48, 31, 13], and are now applied in the scene layout generation area, such as LEGO-Net [37], DiffuScene [33], PhyScene [40], and InstructScene [17]. These methods train diffusion models on layout graphs or bounding boxes to improve physical plausibility. They often model object-object relations coarsely, lack expressiveness for human spatial preferences, and are sensitive to room orientation. Most are trained on small-scale datasets like 3D-Front [9], limiting generalization to diverse layouts. Other existing indoor scene datasets [44, 6, 27] have different data format. Hence they cannot be applied in the scene layout generation task.

**Human preference learning for 2D/3D understanding.** Recent LLM research has increasingly explored human preference with reinforcement learning (RL) to enhance reasoning, as seen in models like OpenAI's o1 [15] and DeepSeek-R1 [11]. In vision tasks, Visual-RFT [19] improves spatial grounding via reward tuning for localization and object detection. Closer to our domain, MetaS-

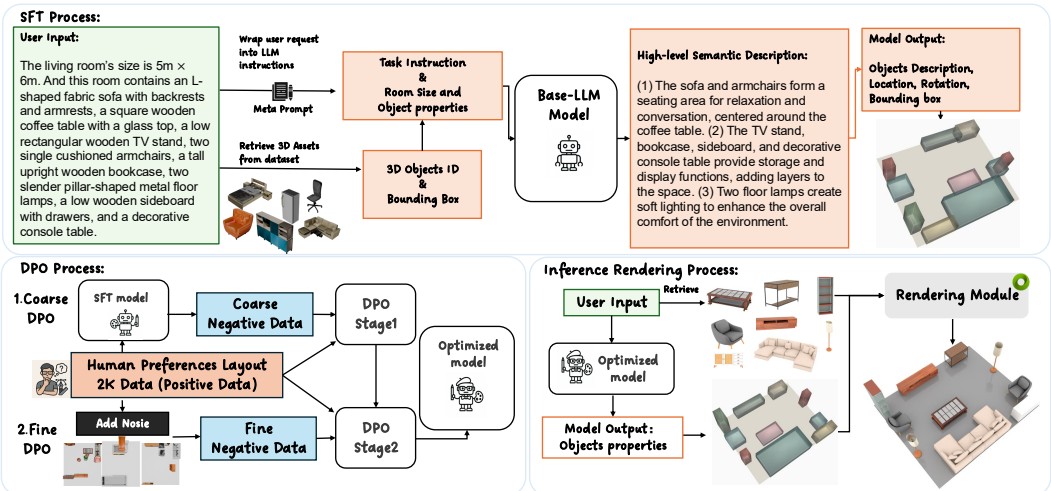

Figure 2: **Overview of the OptiScene pipeline.** (1) **SFT Process**: The user input is transformed into structured instructions and bounding boxes to guide the LLM, which first generates a high-level semantic description before predicting final object placements. (2) **Two-stage DPO**: The model outputs are aligned with human preferences by progressively training on curated positive data and synthesized negative samples with increasing difficulty. (3) **Inference and Rendering**: The optimized model generates layouts that are visualized through a rendering module.

patial [24] applies multi-step RL to reorganize cluttered scenes by relocating objects semantically. While effective for spatial adjustment, it focuses on rearrangement rather than full layout generation. In contrast, we aim to generate complete layouts from scratch, optimizing both structure and usability through preference-aligned method.

# 3 Methodology: 3D-SynthPlace & OptiScene

In this section, we present the 3D-SynthPlace dataset and OptiScene framework (see Fig. 2). We begin by providing a concise formulation of our layout generation problem in Section 3.1. In Section 3.2, we describe the dataset construction process and highlight the criteria and techniques used to ensure high-quality and diverse scene representations. Following this, in Section 3.3, we detail the supervised fine-tuning (SFT) process, including the model's input/output format and the design of the meta instruction prompt. And finally in Section 3.4, we discuss how to use Direct Preference Optimization (DPO) to perform iterative optimization to make the SFT scene layout result align with human preferences.

## 3.1 Problem formulation

We define indoor layout generation as a spatial reasoning task guided by structured language input. Given a user instruction $U$, the system first wraps the natural language instructions into the JSON-structured input with the floor dimensions $F$, the room type $T$, a set of object descriptions $D_{1\sim N}$, and their quantities $Q_{1\sim N}$. The $D_{1\sim N}$ are applied to retrieve the relevant 3D object assets $O = \{o_1, o_2, \ldots, o_N\}$ from a large-scale 3D assets database $DB$ (*eg.* Objaverse [7]), where each object is associated with a corresponding 3D bounding box size. This can be formulated as $\{O, bbox\}_{1\sim N} = R(\{D_{1\sim N}\}, DB)$.

The previously described room and object descriptions are converted into a structured JSON specification $X$. We then apply the meta prompt template of static generation as $P$, which is a fixed text wrapper for sorting room and objects descriptions into a fluent text instruction, and then guiding the LLM $M$ for effective design. The $M$ then generates a layout $L$ from the prompt $P(X)$, specifying 3D positions $c_{1\sim N}$ and orientations $r_{1\sim N}$ in a JSON structure. The $L$ combines the generated objects' properties and input room. Finally, a rendering module $\mathscr{R}_{\text{vis}}$ visualizes the 3D scene layout $V_{\text{img}}$. The whole pipeline is formally defined as:

$$V_{\text{img}} = \mathscr{R}_{\text{vis}}\big(M\big(P(X)\big), O\big), \quad where\{O, bbox\}_{1\sim N} = R\big(\{D_{1\sim N}\}, DB\big). \tag{1}$$

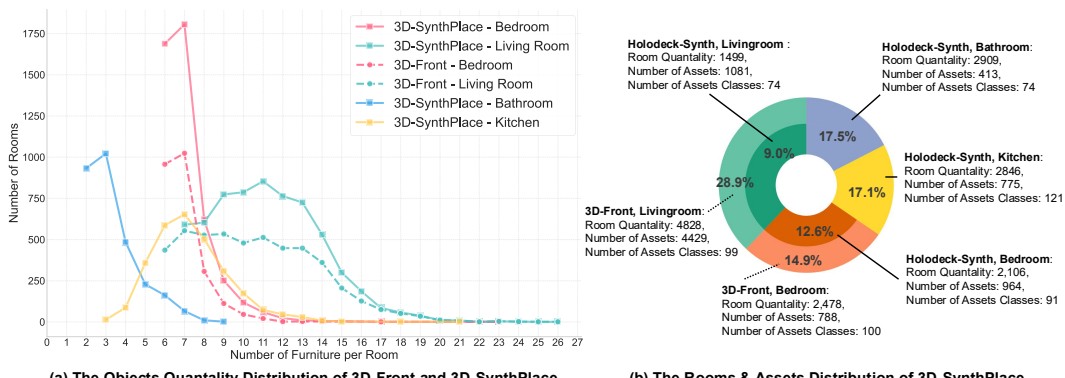

(a) The Objects Quantality Distribution of 3D-Front and 3D-SynthPlace    (b) The Rooms & Assets Distribution of 3D-SynthPlace

Figure 3: **Data distribution analysis of 3D-SynthPlace.**

To support effective training of the layout generation model $M$, we require a large-scale dataset that provides diverse scene configurations and is aligned with human preferences.

## 3.2 3D-SynthPlace dataset construction

While 3D-Front [9] is widely used in layout generation tasks, it suffers from limited scale and scene diversity. After filtering, only around 7,300 layouts are usable, and most belong to bedroom or living/dining room categories. This narrow distribution hinders generalization, especially for underrepresented room types like kitchens and bathrooms. Manually creating large-scale, high-quality layout datasets is prohibitively costly, making synthetic generation a practical alternative. To this end, we construct **3D-SynthPlace**, a large-scale dataset that augments 3D-Front with over 9,000 additional synthetic layouts, enabling richer training coverage for our reasoning- and preference-aligned learning framework.

We use Holodeck [42] as the base generator for constructing the 3D-SynthPlace dataset because manually curating scene layouts is expensive and time-consuming. Holodeck offers a low-cost, easy-to-use alternative by automatically generating layouts from GPT-based prompts. However, during synthesis we observe frequent structural issues: under-populated layouts, clustered object placements in large rooms, and erroneous object counts or orientations. Approximately 55% of the scenes exhibit such flaws. We apply a strict filtering to remove invalid samples and manually correct plausible ones. We also discard scenes with fewer than 6 objects in bedrooms/living rooms, fewer than 3 in kitchens, and fewer than 2 in bathrooms, to ensure minimum complexity. However, Holodeck's default prompts (only consider room description) and output formats differ from our task requirements. To bridge this gap, we (1) unify scene descriptions into a structured JSON input specifying object types and quantities, (2) align asset metadata with 3D-Front conventions, and (3) supplement each layout with a high-level semantic summary describing spatial organization. These semantic descriptions are generated by rendering each layout from top-down and oblique views and prompting GPT to describe spatial relations, providing interpretable supervision for the model's reasoning stage.

The resulting 3D-SynthPlace dataset contains over 16,000 scenes across four room categories, combine with 7,306 3D-Front data samples and 9,360 Holodeck-Synth data samples, balancing geometric coverage with semantic diversity of four room types. Figure 3 compares object counts, room type distributions, and the diversity of newly introduced Objaverse assets between 3D-Front and our dataset. Additionally, we further extend our reasoning step supervision by generating consistent semantic summaries for all data within the 3D-SynthPlace, and annotate retrieved objects with more precise category and geometry metadata. Together, these refinements make 3D-SynthPlace a more comprehensive and reasoning-friendly dataset for indoor layout modeling. For further dataset details and visualizations, please refer to the supplementary material.

## 3.3 Warm-up stage I: supervised fine-tuning (SFT) process

After having constructed a high-quality dataset, we leverage 3D-SynthPlace to activate the spatial reasoning and generation capabilities of existing LLMs. We first adapt a general-purpose LLM to the

layout generation task via supervised fine-tuning (SFT) on the 3D-SynthPlace dataset. Each input instruction $\boldsymbol{X}$ consists of the room type $\boldsymbol{T}$, floor size $\boldsymbol{F}$, object descriptions $\boldsymbol{D}_{1\sim N}$, quantities $\boldsymbol{Q}_{1\sim N}$, and retrieved bounding boxes $\boldsymbol{bbox}_{1\sim N}$, all wrapped into a meta prompt $\boldsymbol{P}$ to condition the model.

To improve spatial reasoning and layout quality, we introduce a **semantic reasoning step** before the final placement prediction. Instead of directly generating object positions and rotations, the model is prompted to produce a high-level natural language summary $\boldsymbol{S}_{\text{sem}}$ describing the global spatial structure (e.g., "place the sofa against the long wall with the coffee table in front"). This step serves as an coarse interpretable intermediate prior, helping the model produce fine coherent layouts and better reflect human design preferences, especially in the absence of explicit spatial constraints.

We treat the LLM as a policy $\pi_\theta$, where $\theta$ denotes the model parameters. This policy generates scene layouts conditioned on input instructions $\boldsymbol{X}$. The full generation process is:

$$\left\{\boldsymbol{X}, \boldsymbol{S}_{\text{sem}}, [\boldsymbol{c}, \boldsymbol{r}]_{1\sim N}\right\} = \pi_\theta\left[\boldsymbol{P}(\{\boldsymbol{D}, \boldsymbol{Q}, \boldsymbol{bbox}\}_{1\sim N}, \boldsymbol{T}, \boldsymbol{F})\right], \tag{2}$$

$$\boldsymbol{L} = (\boldsymbol{T}, \boldsymbol{F}, \{\boldsymbol{D}, \boldsymbol{Q}, \boldsymbol{bbox}, \boldsymbol{c}, \boldsymbol{r}\}_{1\sim N}). \tag{3}$$

We minimize the discrepancy between predictions and ground-truth layouts, including both the semantic summary and structured output. The prompt includes design constraints to (1) avoid collisions, (2) maintain functional organization, and (3) favor boundary-aligned placement to enhance spatial openness. Full prompt templates are provided in the supplementary material.

### 3.4 Reinforcing stage II: human preference optimization

While our supervised fine-tuning (SFT) enables the model to acquire structured spatial priors, we observe that it still struggles to generate physically valid and semantically coherent layouts. In particular, generated scenes often suffer from object collisions, boundary violations, or even high-level planning inconsistencies due to hallucinations in the reasoning process. We attribute these issues to the inherent limitations of LLMs in modeling 3D spatial structures, where the learned relations between objects tend to remain coarse, linguistic, and geometry-agnostic, rather than grounded in physical feasibility or human design preferences. To address this gap, we propose a multi-stage optimization framework based on **Direct Preference Optimization (DPO)** that progressively aligns the model outputs with human preferences and improves its understanding of 3D scene plausibility.

The DPO framework is formulated as a preference modeling problem, where the model learns to favor human-preferred layouts $\boldsymbol{L}^+$ over less desirable alternatives $\boldsymbol{L}^-$ given the same input $\boldsymbol{X}$ from the 3D-SynthPlace dataset $\mathscr{D}$. Formally, we optimize the policy $\pi_\theta$ using the standard DPO loss:

$$\max_{\pi_\theta} \mathbb{E}_{\left(\boldsymbol{X}_i, \boldsymbol{L}_i^+, \boldsymbol{L}_i^-\right)\sim\mathscr{D}} \log \sigma\left(\beta \log \frac{\pi_\theta\left(\boldsymbol{L}_i^+ \mid \boldsymbol{X}_i\right)}{\pi_{\text{ref}}\left(\boldsymbol{L}_i^+ \mid \boldsymbol{X}_i\right)} - \beta \log \frac{\pi_\theta\left(\boldsymbol{L}_i^- \mid \boldsymbol{X}_i\right)}{\pi_{\text{ref}}\left(\boldsymbol{L}_i^- \mid \boldsymbol{X}_i\right)}\right), \tag{4}$$

where $\beta$ is a temperature hyperparameter and $\sigma(\cdot)$ is the sigmoid function. Due to the complexity of layout generation and the nuanced nature of human design preferences, we adopt a two-stage DPO framework to progressively refine the SFT model.

**DPO-stage I: human consensus alignment.** Instead of relying on noisy or random model generations to create preference pairs, we take a data-first approach. We observe that curated layouts from 3D-SynthPlace inherently reflect strong human preferences in object placement and room organization. We select $\sim 2{,}200$ of such layouts that received unanimous approval from annotators as positive samples $\boldsymbol{L}^+$. For each corresponding scene description $\boldsymbol{X}$, we then use our SFT model to generate multiple layout candidates. These generated results, while semantically plausible, often deviate from human-intuitive arrangements, and are therefore treated as negative samples $\boldsymbol{L}^-$. This contrastive pairing ensures that the optimization target explicitly favours human consensus without being biased by generation-stage noise.

**DPO-stage II: geometric plausibility enhancement.** While the first-stage DPO improves layout semantics and global structure, it does not always eliminate low-level physical errors, such as object collisions, surface overflow, or implausible rotations. These issues arise because the model lacks direct training signals to avoid physically invalid configurations. To address this, we design a second optimization stage focused on hard negative mining: for each ground-truth layout $\boldsymbol{L}^+$, we synthesize perturbed versions $\boldsymbol{L}^-$ with targeted spatial violations (e.g., objects extending beyond table bounds, interpenetrations with walls or other furniture). These negatives are subtle but structurally invalid, providing sharper gradients to improve the model's sensitivity to physical realism.

Table 1: Layout quality [FID ↓ / OOR ↓] on 3D-Front & 3D-SynthPlace.

| Method | 3D-Front | | 3D-SynthPlace | | | |
| --- | --- | --- | --- | --- | --- | --- |
| | Bedroom | Living room | Bedroom | Living room | Kitchen | Bathroom |
| I-Design [2] | 73.56 / 0.19 | 80.123 / 0.20 | 76.42 / 0.18 | 80.561 / 0.19 | 90.25 / 0.28 | 85.4 / 0.16 |
| DiffuScene [33] | 49.71 / 0.12 | 55.21 / 0.14 | 58.37 / 0.14 | 61.46 / 0.18 | - / - | - / - |
| LLplace [41] | 56.63 / 0.12 | 61.97 / 0.17 | 63.45 / 0.06 | 69.64 / 0.16 | 89.34 / 0.14 | 59.31 / 0.03 |
| DiffuScene (w/ 3D-Synth) [33] | 41.23 / 0.06 | 46.64 / 0.11 | 38.96 / 0.06 | 44.25 / 0.12 | 52.63 / 0.10 | 39.76 / 0.04 |
| LLplace (w/ 3D-Synth) [41] | 38.81 / 0.07 | 45.32 / 0.13 | 37.66 / 0.04 | 40.72 / 0.08 | 43.68 / 0.08 | 37.44 / 0.03 |
| OptiScene | **33.56 / 0.03** | **37.68 / 0.08** | **26.45 / 0.01** | **31.25 / 0.02** | **41.73 / 0.05** | **35.29 / 0.02** |

Table 2: Layout quality using GPT-4o[Func ↑ / Layout ↑ /Aes. ↑] on 3D-Front & 3D-SynthPlace.

| Method | 3D-Front | | 3D-SynthPlace | | | |
| --- | --- | --- | --- | --- | --- | --- |
| | Bedroom | Living room | Bedroom | Living room | Kitchen | Bathroom |
| I-Design [2] | 6.7 / 7.0 / 7.3 | 6.5 / 6.4 / 7.1 | 6.4 / 6.2 / 7.0 | 6.1 / 6.0 / 6.8 | 5.8 / 5.6 / 6.6 | 6.0 / 6.1 / 6.9 |
| DiffuScene [33] | 7.8 / 8.1 / 8.4 | 7.5 / 7.9 / 8.2 | 8.0 / 8.2 / 8.5 | 7.9 / 7.8 / 8.3 | 8.2 / 8.0 / 7.8 | 7.4 / 7.6 / 8.0 |
| LLplace [41] | 8.6 / 8.4 / 8.5 | 8.2 / 7.8 / 8.0 | 8.5 / 8.6 / 8.8 | 7.2 / 7.0 / 8.2 | 7.4 / 7.8 / 8.0 | 7.5 / 7.3 / 8.0 |
| **OptiScene** | **8.6 / 9.1 / 9.3** | **8.8 / 8.7 / 9.0** | **8.7 / 9.0 / 9.2** | **8.9 / 9.2 / 9.1** | **8.4 / 8.3 / 8.8** | **8.7 / 9.4 / 9.0** |

Through this two-stage optimization, our model **OptiScene** develops dual awareness of both design intent and spatial plausibility. Unlike generic LLMs or prompt-based tools, OptiScene is explicitly trained to meet the needs of interior design tasks, producing layouts that are not only spatially valid but also aligned with real-world usage and human preferences.

## 4 Experiments

### 4.1 Experiments setup

**Dataset setup.** Our constructed dataset, **3D-SynthScene**, contains a total of 16,666 scenes, including 7,306 original layouts from the 3D-Front dataset and 9,360 synthetic layouts generated by Holodeck-Synth pipeline. Following the evaluation protocol of LayoutGPT [8] and LLplace [41], we select 423 bedroom and 53 living room layouts from 3D-Front as the test set. Additionally, we sample 50 layouts for each of the four room types—bedroom, living room, kitchen, and bathroom—from Holodeck-Synth as an extended test set. The remaining 15,990 scene layouts are used as the training set. As described in Section 3.4, we manually select 2,200 high-quality human-preferred layouts from the full set of 15,990 training samples as the positive dataset for DPO.

**Training setup.** We adopt Qwen3-8B as the base LLM for both supervised fine-tuning (SFT) and direct preference optimization (DPO). Due to resource constraints, we apply LoRA-based parameter-efficient fine-tuning. For SFT, we set the LoRA $\alpha$ to 32, rank $r$ to 16, and dropout rate to 0.05. We train the model for 10 epochs using a learning rate of $5 \times 10^{-6}$ with a cosine learning rate scheduler. For the two-stage DPO, we use the same LoRA configuration but lower the learning rate to $5 \times 10^{-7}$ and train for 5 epochs with the same cosine scheduler. The whole training process takes roughly 140 GPU hours.

**Evaluation metrics.** We adopt four metrics to comprehensively evaluate layout quality. (1) **FID**, following DiffuScene [33], measures visual realism between rendered and real scenes. (2) **Object Overlap Rate (OOR)** evaluates physical plausibility by computing the proportion of intersecting bounding boxes. (3) **GPT-4o Ratings** assess layout quality from three dimensions—*Functionality*, *Furniture Layout*, and *Aesthetics*—scored from 0 to 10 via prompt-based evaluation. (4) **Usability Rate (UR)** reflects human acceptability: among 50 sampled scenes, each judged by 5 annotators, a layout is counted as usable if at least 3 consider it physically and semantically valid.

### 4.2 Experiment results

**FID & OOR metrics.** Table 1 reports layout quality across two metrics: FID (↓) and OOR (↓). Overall, models trained with the extended 3D-SynthPlace data show significant improvements over their 3D-Front-only

Table 3: Layout generation success rate.

| Method | Bedroom | Living room | Kitchen | Bathroom |
| --- | --- | --- | --- | --- |
| DiffuScene (w/ 3D-Synth) [33] | 30% | 20% | 20% | 25% |
| LLplace (w/ 3D-Synth) [41] | 40% | 30% | 20% | 35% |
| Holodeck [42] | 55% | 30% | 25% | 35% |
| **OptiScene** | **75%** | **50%** | **30%** | **45%** |

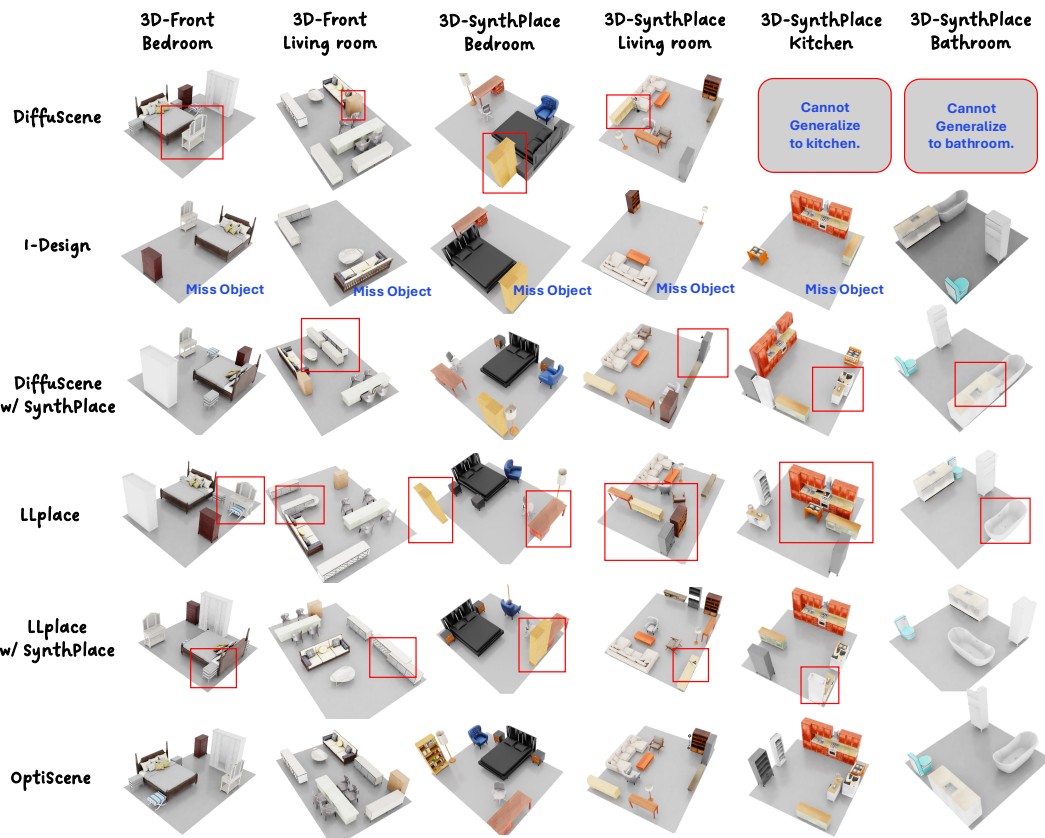

Figure 4: **Sample layouts generated by several models on 3D-Front & 3D-SynthPlace.** Common layout errors—such as misaligned objects, or object collisions, are highlighted with red boxes. (Please zoom in to see the details.)

counterparts—demonstrating the effectiveness of large-scale, diverse training layouts. Among all methods, OptiScene achieves the lowest FID and OOR across every room type, including newly introduced categories such as kitchen and bathroom. Compared to LLplace (w/ 3D-Synth) in 3D-SynthPlace, OptiScene reduces FID by 11.21 in bedrooms and 9.47 in living rooms, while also halving the OOR in several cases.

**GPT-Based evaluation.** We further evaluate layout quality using GPT-4o across three aspects: functionality, layout rationality, and aesthetics. As shown in Table 2, OptiScene consistently achieves the highest scores across all room types and metrics. On the 3D-SynthPlace test set, it surpasses DiffuScene and LLplace by an average margin of +0.9 in functionality and +1.0 in rationality. In more challenging scenes such as kitchens and bathrooms, OptiScene reaches up to 9.4 in rationality, reflecting stronger spatial reasoning and layout usability.

**Usability consistency.** To evaluate the robustness and reliability of different layout generation methods, we conduct a usability consistency test in which each model generates 50 layouts per room type, and we record the proportion of layouts deemed valid (i.e., free of collisions, interpenetrations, and major structural violations). As shown in Table 3, OptiScene significantly outperforms all baselines in both bedroom and living room categories, achieving 75% and 50% usable layouts respectively. In contrast, DiffuScene and LLplace achieve much lower success rates, especially on complex scenes such as kitchens and bathrooms. While Holodeck performs reasonably well in bedrooms, its outputs remain less stable across room types.

**Qualitative analysis.** Figure 4 shows visual comparisons across models on six room categories from 3D-Front and 3D-SynthPlace. I-Design consistently fails to construct scenes with more than 5–6 objects, often missing essential furniture (marked in blue), highlighting its limited scalability. DiffuScene and LLplace trained only on 3D-Front struggle to generalize, especially to novel room

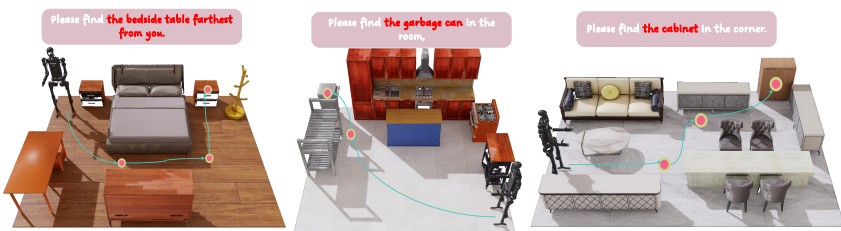

Figure 5: **Generated layout for robot object-centric navigation.** The figure shows three generated scenes for the instruction-based robot navigation.

Table 4: Room editing performance.

| Method | OOR ↓ | ESR ↑ |
|---|---|---|
| OptiScene | 0.023 | - |
| OptiScene-Editing | 0.028 | 90% |

Table 5: Object-centric navigation performance.

| LLM | SR ↑ | NE ↑ |
|---|---|---|
| GPT-4o | 87.56 | 1.00 |
| Qwen-3-8B | 75.92 | 0.98 |
| Chat-GLM4-flash | 72.33 | 0.92 |

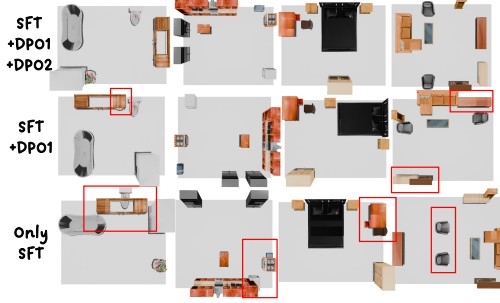

Figure 6: Progressive improvement over stages.

types like kitchens and bathrooms. DiffuScene in particular fails entirely in these categories. With 3D-SynthPlace training, both methods see noticeable improvement, yet issues like collisions and misalignments (red boxes) persist. In contrast, our proposed OptiScene generates coherent, well-structured layouts across all scenes, with correct functional grouping and minimal physical violations, demonstrating superior generalization and human-aligned spatial reasoning.

## 4.3 Downstream tasks

**Room editing** is essential for interactive layout generation, yet prior methods like DiffuScene and I-Design lack such capability. Inspired by LLplace, we extend 3D-SynthPlace with an object-level editing dataset containing addition and removal instructions. We fine-tune the model with mixed editing and original data, and evaluate on 20 edit cases using object-overlap rate (OOR) and editing success rate (ESR). As shown in Table 4, OptiScene maintains low OOR post-editing and achieves 90% ESR, with only two failed cases.

**Object-Centric navigation** evaluates whether the generated layouts support goal-directed robot exploration. Given a command like "Find a *<object>* in the room," the robot succeeds if the target appears within ±30° of its field of view and within 2 meters, measured by success rate (SR). We also report navigation error (NE), defined as the final distance to the target. As shown in Table 5, our layouts support stable navigation across LLM backends, indicating strong spatial usability. We show some robot navigation results in our generated 3D scenes in Fig. 5.

## 4.4 Ablation study

**Effect of high-level semantic reasoning.**
We investigate whether prompting the model to generate high-level spatial semantics before layout prediction improves overall quality. As shown in Table 6, adding this reasoning step to the SFT model increases layout usability (from 28% to 33%). This demonstrates that explicitly guiding the model to

Table 6: Ablation Study of OptiScene.

| Only SFT | Reasoning | SFT+DPO1 | SFT+DPO1+DPO2 | OOR. | UR. |
|---|---|---|---|---|---|
| ✓ | - | - | - | 0.049 | 28% |
| ✓ | ✓ | - | - | 0.048 | 33% |
| ✓ | ✓ | ✓ | - | 0.033 | 40% |
| ✓ | ✓ | ✓ | ✓ | **0.023** | **50%** |

first consider object relationships and room intent leads to more coherent and human-aligned scene structures.

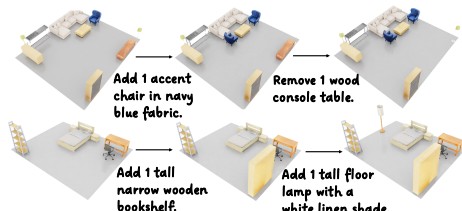
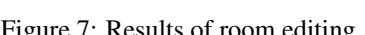

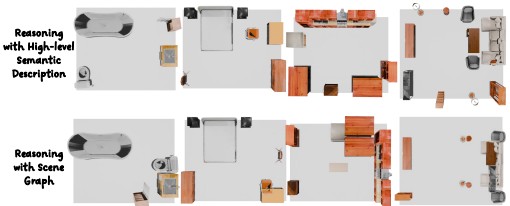

Figure 7: Results of room editing.

Figure 8: Comparison between high-level semantic reasoning & scene graph reasoning.

**Ablation study on SFT and Two-Stage DPO.**   As shown in Table 6, adding reasoning improves OOR and UR by encouraging coherent object planning. DPO-Stage1 aligns outputs with human preferences, further reducing violations, while DPO-Stage2 introduces targeted noise to correct fine-grained physical errors. Qualitative results in Fig. 6 illustrate progressive improvement in layout realism and usability across stages.

**High-level semantic reasoning v.s. scene graph reasoning.**   We compare two reasoning supervision strategies: high-level semantic descriptions and scene graphs. As shown in Fig. 8, using natural language descriptions leads to significantly better layout quality, with more coherent and functional arrangements. This is because LLMs are inherently better at interpreting and generating natural language than symbolic graph structures. Scene graphs, while explicit, lack global spatial context and are less aligned with the model's pretraining distribution, resulting in fragmented or rigid layouts.

## 5    Conclusion

In this paper, we present **OptiScene**, a human-aligned indoor layout generation framework based on a fine-tuned open-source LLM. To overcome limitations of existing approaches—such as poor spatial understanding, lack of control, and limited scalability—we construct **3D-SynthPlace**, a high-quality dataset that integrates manually refined layouts synthesized by Holodeck with the original 3D-Front scenes. Built on this data, we introduce a novel supervised fine-tuning (SFT) strategy that incorporates a high-level semantic reasoning step before final layout prediction, enabling the model to internalize human-aligned spatial priors. Furthermore, we propose a two-stage Direct Preference Optimization (DPO) pipeline to refine the model's outputs based on human-preferred layouts and physical plausibility constraints. Extensive results show that OptiScene surpasses prior baselines in layout quality and usability. It also generalizes well to downstream tasks such as scene editing and robot navigation. These findings underscore the potential of aligning LLMs with structured reasoning and human preferences for controllable and deployable layout generation.

**Limitation and future work**   Our work has some limitations that can be improved in future work: 1. We focus on the generation of a single room layout, without considering the generation of multiple room layouts. Our future work focuses on generating the layout of an entire house or building. 2. Our a scene dataset considers only the furniture. Our future work focuses on collecting more small objects and wall objects to build a more complete and detailed room layout, further enhancing the authenticity of the room.

## Acknowledgments

We thank all anonymous reviewers for their constructive reviews, and the support from all authors' institutions.

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

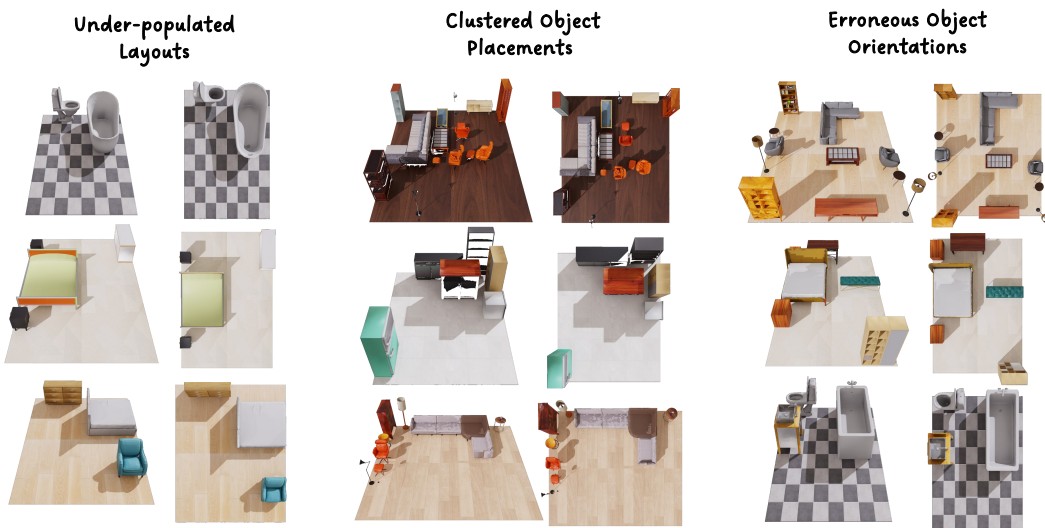

Figure 9: The deleted error data which generated by Holodeck.

## A More details of 3D-SynthPlace dataset

### A.1 3D scene layout alignment

In the Holodeck [42] generation data, (0, 0) is generally used as the bottom-left corner of the floor, while the single room information in 3D-Front is extracted from a large room formed by a collection of multiple rooms, making the floor center more uncertain. To align the centers of the rooms in both datasets, we use (0, 0) as the floor center. We calculate the geometric center of the floor based on the floor boundary coordinates of both datasets and translate the floor boundary and the coordinates of objects within the room accordingly. The offset is the difference between (0, 0) and the original geometric center of the floor. Additionally, we find that although both Holodeck and 3D-Front use a $y$-axis coordinate system, their $xz$-axis rotation directions are opposite. To unify the rotation directions, we standardize to the 3D-Front rotation direction by adding 180° to the rotation angle of objects in the Holodeck, thus making the rotation angles consistent.

### A.2 Deleted Holodeck generated data

As discussed in Section 3.2, the 3D scene layout rooms created by the Holodeck generation process have a low success rate. We filter the data and delete badly generated data based on three issues: (1) under-populated layouts, (2) clustered objects placed in large rooms, and (3) erroneous object counts or orientations (see Fig. 9).

**Under-populated Layouts**: These layouts have a noticeably insufficient number of objects, making the space appear sparse and unnatural. Particularly in bathrooms and bedrooms, the lack of sufficient furniture and decorations results in a lack of liveliness and functionality. Such layouts may not effectively simulate real-life environments, impacting the model's training effectiveness.

**Clustered Object Placed in Large Rooms**: In large rooms, objects are placed too closely together, leading to uneven space utilization. This can cause visual disharmony and fail to realistically reflect the reasonable distribution of furniture in real life. This issue is especially pronounced in living rooms and kitchens, affecting both functionality and aesthetics.

**Erroneous Object Counts or Orientations**: Errors in the number or orientation of objects lead to unreasonable room layouts. For example, the orientation of furniture may not match the room's structure, or the number of objects may be too high or too low, affecting the overall harmony of the room. Such errors can mislead the model in learning spatial relationships, impacting reasoning capabilities.

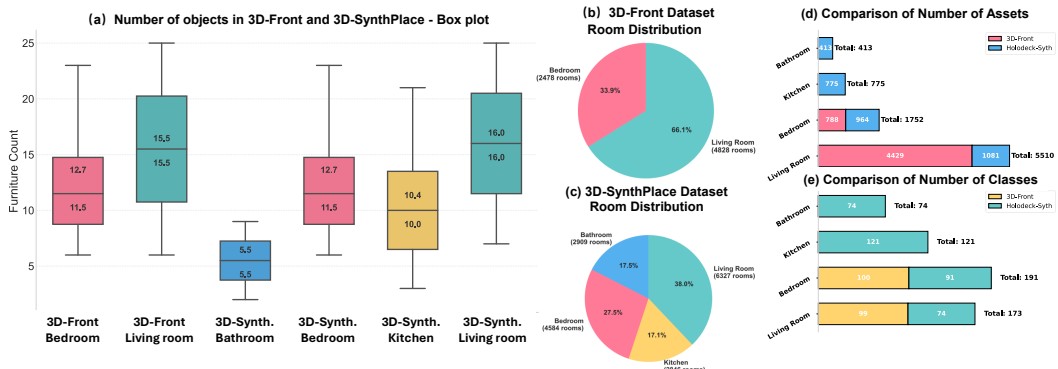

Figure 10: Comparison of object counts, room type distributions, and asset/class statistics between the 3D-Front and 3D-SynthPlace datasets.

In addition to filtering out invalid layouts, we exclude small objects produced during the Holodeck generation process, as they are not compatible with the problem definition of our room layout formulation.

### A.3 Data distribution description

Figure 10 presents a comprehensive distribution comparison between the 3D-Front and 3D-SynthPlace datasets across various dimensions:

**Number of Objects, Fig. 10(a)**: The box plot illustrates the distribution of furniture counts in different room types for both datasets. The 3D-Front dataset shows a higher median count in living rooms compared to bedrooms, while the 3D-SynthPlace dataset exhibits a similar pattern with slightly higher counts in living rooms and kitchens.

**Room Distribution, Fig. 10(b & c)**: Pie charts depict the room distribution within each dataset. The 3D-Front dataset is predominantly composed of living rooms (66.1%), followed by bedrooms (33.9%). In contrast, the 3D-SynthPlace dataset has a more diverse distribution, with living rooms (38%) and bedrooms (27.5%) being the most common, alongside a significant portion of bathrooms (17.5%) and kitchens (17.1%).

**Comparison of Number of Assets, Fig. 10(d)**: The bar chart compares the total number of assets across different room types. The 3D-Front dataset has a significantly higher total number of assets, especially in living rooms, compared to the Holodeck-Synth dataset.

**Comparison of Number of Classes Fig. 10(e)**: The bar chart compares the number of classes available in each dataset. The 3D-Front dataset generally has more classes across different room types, with the most notable difference in the living room category.

### A.4 High-level semantic descriptions and reasoning

Figure 11 shows examples of high-level semantic descriptions and reasoning with the 3D scene layouts. Examples of generated room layouts paired with natural language descriptions. Each row shows two 3D layout visualizations of the same room from different angles, accompanied by a GPT-generated description of the spatial arrangement and object relations. The descriptions emphasize functionality, object alignment, and spatial reasoning without relying on directional terms like "left" or "right".

## B   Instruction-to-JSON Wrapper

We use a Qwen2.5-7B-Instruct LLM as a wrapper to convert natural language user instructions into structured JSON. The process works as follows:

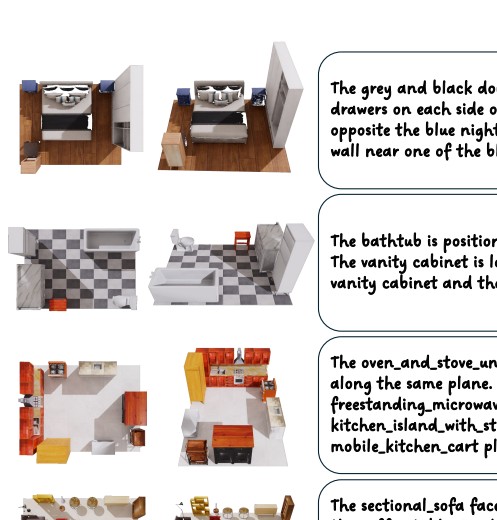

The grey and black double bed is centrally positioned in the room, with a blue nightstand with drawers on each side of the bed. The wooden corner side table stands near one side of the bed, opposite the blue nightstands. The grey wardrobe with shelves and drawers is placed against the wall near one of the blue nightstands.

The bathtub is positioned next to the storage cabinet. The toilet is placed opposite the bathtub. The vanity cabinet is located adjacent to the step stool, with the step stool placed between the vanity cabinet and the toilet.

The oven_and_stove_unit is positioned adjacent to the kitchen_cabinet_base, which extends along the same plane. The tall_pantry_cabinet stands near the kitchen_cabinet_base, with the freestanding_microwave_cabinet located across an open space from both. The kitchen_island_with_storage is situated opposite the tall_pantry_cabinet, with the mobile_kitchen_cart placed near the kitchen_island_with_storage.

The sectional_sofa faces the coffee_table, with two armchairs positioned on the opposite side of the coffee_table. A side_table and floor_lamp are placed near one end of the sectional_sofa, while two additional side_tables and another floor_lamp are situated close to the armchairs. The tv_stand and bookshelf are against the far wall, and the sideboard is placed adjacent to the bookshelf.

The sectional_sofa is positioned opposite the tv_stand, with the coffee_table placed centrally between them. Two armchairs are facing the sectional_sofa, each adjacent to the coffee_table. The bookshelf and sideboard are placed along the walls, with a floor_lamp positioned near the armchairs and another floor_lamp closer to the tv_stand.

Figure 11: High-level semantic descriptions and reasoning with the corresponding 3D scene layouts.

- The wrapper extracts room size, room type, object types, and object counts from the instruction.

- If any field is missing, the wrapper heuristically infers it based on existing information (e.g., number and type of objects).

- It then retrieves suitable 3D assets for each object. We follow a purely text-based retrieval approach, inspired by Holodeck [42], to match the objects mentioned in the user instruction to assets in our 3D object database.

- The extracted values are inserted into a predefined JSON layout template, which is then used as part of the prompt input to OptiScene for inference. This wrapper provides robustness by normalizing diverse instructions, modularity for easier debugging, and flexibility through noise filtering.

## C    More details of SFT

Section 3.3 describes SFT. Here, we discuss the details of the meta instruction prompt. First, we illustrate the prompt template in the Table 7 and Table 8. As shown in the tables, to fine-tune the model for the 3D room layout generation task, we design a structured meta prompt that sets the model as a skilled room layout designer. The prompt guides the model through a step-by-step reasoning and generation process, covering object extraction, spatial analysis, layout planning, and final output formatting. It explicitly incorporates design heuristics such as edge-aligned placement, alignment to walls, and functional constraints (e.g., chairs must face desks). The model is instructed to reason about object relationships without using explicit directions (e.g., "left", "right") and to output both the reasoning process and the final layout in a well-defined JSON format. The response is enclosed in a structured JSON-like format for better consistency and parsing. The prompt also includes post-checklists to ensure validity of the generated layout (e.g., no overlap, correct coordinate units, functional flow). This design ensures that the model not only generates spatially valid scenes but also explains its decisions in a transparent and interpretable way.

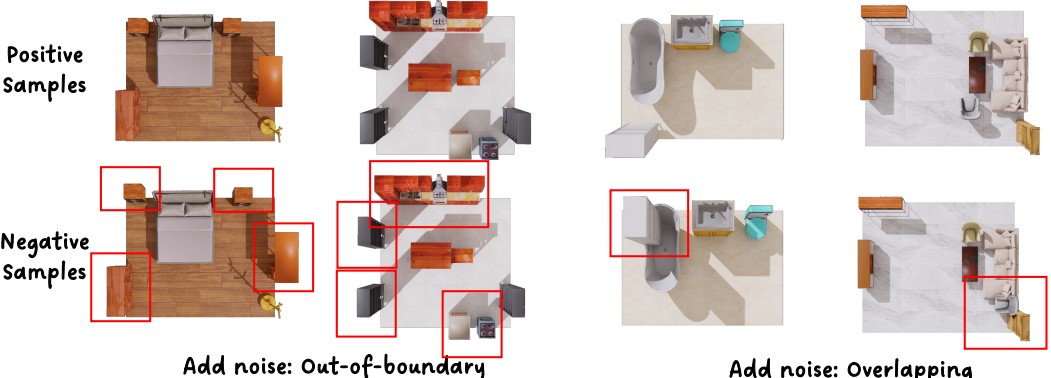

Figure 12: Some positive scene layouts and negative scene layouts with noise additions in the DPO-stage2.

# D  More details of DPO

Section 3.4 introduces the two-stage DPO framework designed to align generated room layouts with human preferences. We begin by selecting 1,100 scene layouts from the Holodeck-Synth dataset and another 1,100 from the 3D-Front dataset. These layouts are characterized by reasonable, functional, and human-aligned spatial arrangements. In the second stage of DPO training (DPO-stage 2), we generate hard negative samples by adding targeted spatial violations to the positive examples. Specifically, as illustrated in Fig. 12, we introduce two types of noise: (1) Out-of-boundary violations, where furniture items extend beyond the floor area; and (2) Object-overlap violations, where furniture pieces intersect or collide with each other. Despite improvements from SFT and DPO-stage 1, the model still occasionally produces such errors. Therefore, we further fine-tune the model using these challenging negative samples to enhance spatial reasoning and layout robustness.

# E  More details of the experiments

## E.1  Inference and rendering setup

All layout generation results are produced using the optimized model obtained after the second stage of DPO. To ensure physically plausible and visually accurate visualization of scenes, we employ **NVIDIA Isaac Sim** [22] as the rendering and simulation backend.

## E.2  Details of the evaluation metrics

**Object overlap rate (OOR).** This metric quantifies the spatial overlap between a set of 2D bounding boxes. To calculate the OOR, we first extract the position and size information from each object, where the position is typically represented as $(x, y)$ and the size is given by width and depth. Using this information, we create 2D bounding boxes with the center at the object's position and dimensions determined by the size. We then compute the area of each bounding box and, for each pair of bounding boxes, we calculate the area of their intersection. We sum the intersection areas of all object pairs to obtain the total intersection area, and then sum the areas of all objects to get the total area. The OOR is then defined as the total intersection area divided by the total area:

$$\text{OOR} = \frac{\sum_{i=1}^{N} \sum_{j=i+1}^{N} \text{Area}(\text{Intersection}(B_i, B_j))}{\sum_{i=1}^{N} \text{Area}(B_i)},$$

where $B_i$ represents the bounding box of the $i$-th object, Intersection$(B_i, B_j)$ denotes the intersection area of two bounding boxes, and Area$(B_i)$ is the area of the bounding box.

**GPT-4o evaluation.** Following a similar approach to the evaluation protocol introduced in I-Design [2], we employ GPT-4o as an automated evaluator to assess the quality of our generated room

layouts. Specifically, GPT-4o is prompted to assign scores (ranging from 0 to 10) based on multiple criteria, including functionality, spatial arrangement, and aesthetic coherence, in alignment with the given user preferences. The detailed prompt template used for this evaluation is provided in Table 9.

### E.3    More qualitative results.

Figures 13 and 14 show additional qualitative comparisons between the outputs of our OptiScene model (left two columns) and the ground truth (right two columns) across various room configurations. It is important to emphasize that the objective of OptiScene is not to converge to the ground truth layouts exactly, but rather to generate plausible and functional indoor scenes based on high-level semantic and spatial constraints. The generated layouts exhibit a more efficient use of space, better object alignment, and clearer functional zoning compared to the ground truth. These examples demonstrate the model's capacity to generalize and produce reasonable, sometimes even preferable, alternative layouts that remain faithful to the intended room semantics. This highlights the potential of OptiScene to support diverse and creative room layout configurations.

### E.4    More downstream tasks results.

Figures 15 illustrate more downstream tasks results with room editing. As shown in these figures, with appropriate instructions, the model can edit the room layout.

Table 7: Meta Prompt Template for generation task (Part 1).

**Meta Prompt Template for generation task (Part 1)**

You are a skilled room layout designer. Your task is to arrange [Objects] within a given [Room Type] effectively. Follow these guidance to complete your design:

(1) Extract the [Room Type], [Room Area], [Objects], and [Bounding Box Size] from the provided JSON data. (2) Analyze the spatial relationships among [Objects] within the specified [Room Type]. Pay special attention to **avoiding overlap** and **consider other spatial factors like accessibility and aesthetics**.

(3) Determine and design the precise location of all [Objects] ensuring that their bounding boxes do not overlap and that the layout is functional and visually appealing.

(4) I prefer objects to be placed at the edge (the most important constraint) of the room if possible which makes the room look more spacious.

(5) The objects are usually aligned in some ways (parallel or perpendicular to walls).

(6) Chairs must be placed near to the table/desk and face to the table/desk.

(7) Before specifying the detailed positions of each object, first reason step-by-step about their general arrangement and relative spatial relationships: a) Which objects need the most space or have fixed positions (like beds, wardrobes) b) Which objects need to be grouped together (like nightstands with bed) c) Traffic flow and accessibility considerations. Then, clearly articulate your reasoning process. Emphasize the spatial relationships between objects without using explicit directional terms like "left," "right," "front," or "back." Summarize the overall arrangement in a logical and natural manner, ensuring that all major objects are accounted for.

(8) After presenting the thought process, report your design with detailed 3D space coordinates and rotation angles for each object in JSON format, as follows:

```
    "object": "object",
    "coordinates": [
      {
        "x": x,
        "y": y,
        "z": z
      }
    ],
    "rotate":[
      {
        "angle": r
      }
    ]
}
```

The centroid of the room is {"x": 0.00, "y": 0.00, "z": 0.00"}.
Important Notes about Coordinate System:
- Z-axis points upward (z=0 is floor level)
- Rotation angles are in radians, measured in the XY-plane

Table 8: Meta Prompt Template for generation task (Part 2).

**Meta Prompt Template for generation task (Part 2)**

(9) The response should follow the following format:

```
<reasoning>
[Reason]
...
[/Reason]
</reasoning>
<answer>
[Design]
...
[/Design]
</answer>
```

First carefully read this example:

```
[Example Room Type]
Bedroom
[/Example Room Type]

[Example Objects and Bounding Box Size]
/* A fixed example is put here to show the input format*/
[/Example Objects and Bounding Box Size]

[Example Reason]
/* A fixed example is put here to show the reason format*/
[/Example Reason]

[Example Output]
/* A fixed example is put here to show the output format*/
[/Example Output]
```

Now, please proceed with the design task as outlined and provide only the JSON formatted output of your design:

```
[Task Room Type]
/*Input room type*/
[/Task Room Type]

[Task Objects & Bounding Box Size]
/* The JSON format input of objects description
and bounding box size*/
[/Task Objects & Bounding Box Size]
```

Note: the units for the coordinates are meters.
Before submitting your final design, please verify:
- All objects are within room boundaries
- No objects overlap
- Sufficient clearance space exists around furniture
- The layout is practical and functional
- All rotations are properly specified in radians
Now, please proceed with the design task as outlined and provide your thought process and the JSON formatted output of your design:

Table 9: GPT-4o Prompt Template for Room Layout Evaluation.

**GPT-4o Prompt Template for Room Layout Evaluation**

You are an expert in interior design and human-centric spatial planning. Your task is to evaluate the quality of the following room layout renders based on how well they match the user's design preferences, which are provided below (in triple backquotes).

Please assign a numerical score from **0 to 10** (0 = completely inconsistent, 10 = perfectly aligned) considering the following three aspects:

**1. Functionality and Activity-based Alignment**
- Does the layout support natural and efficient use of the space for daily activities (e.g., sleeping, working, relaxing, walking)?
- Are key object groupings (e.g., desk and chair, bed and nightstand) placed functionally and accessibly?
- Is there sufficient circulation space to ensure human accessibility?

**2. Layout and Furniture Placement**
- Are the furniture pieces arranged in a logical, practical way within the room boundaries?
- Are objects positioned with respect to common design principles (e.g., not blocking windows or doors, aligned to walls where appropriate)?
- Are there overlaps or unnatural collisions between objects?

**3. Aesthetic Coherence**
- Is the overall layout visually balanced and spacious?
- Does the arrangement exhibit good proportions, symmetry or asymmetry, and grouping where needed?
- Is the furniture distribution harmonious and pleasing to the eye, according to the user's stated aesthetic preferences?

**User Preferences:**
```{Insert user preferences here.}```
After considering all the above, return your evaluation in the following JSON format:

```
{
```{
 "functionality_score": X,
"layout_score": Y,
 "aesthetics_score": Z,
 "overall_score": M,
"comments": "Brief explanation of your judgment."
 }```
}
```

**Scoring Guidelines:**
- Scores should be integers from 0 to 10.
- The overall score can be the average or holistic assessment across the three criteria.
- Include a brief justification of your scores in the "comments" field (1–3 sentences).
**Note:** The goal is to measure how well the generated layout adheres to functional, spatial, and aesthetic expectations given the user's input. Be fair and critical.

Front         Top         GT-Front         GT-Top

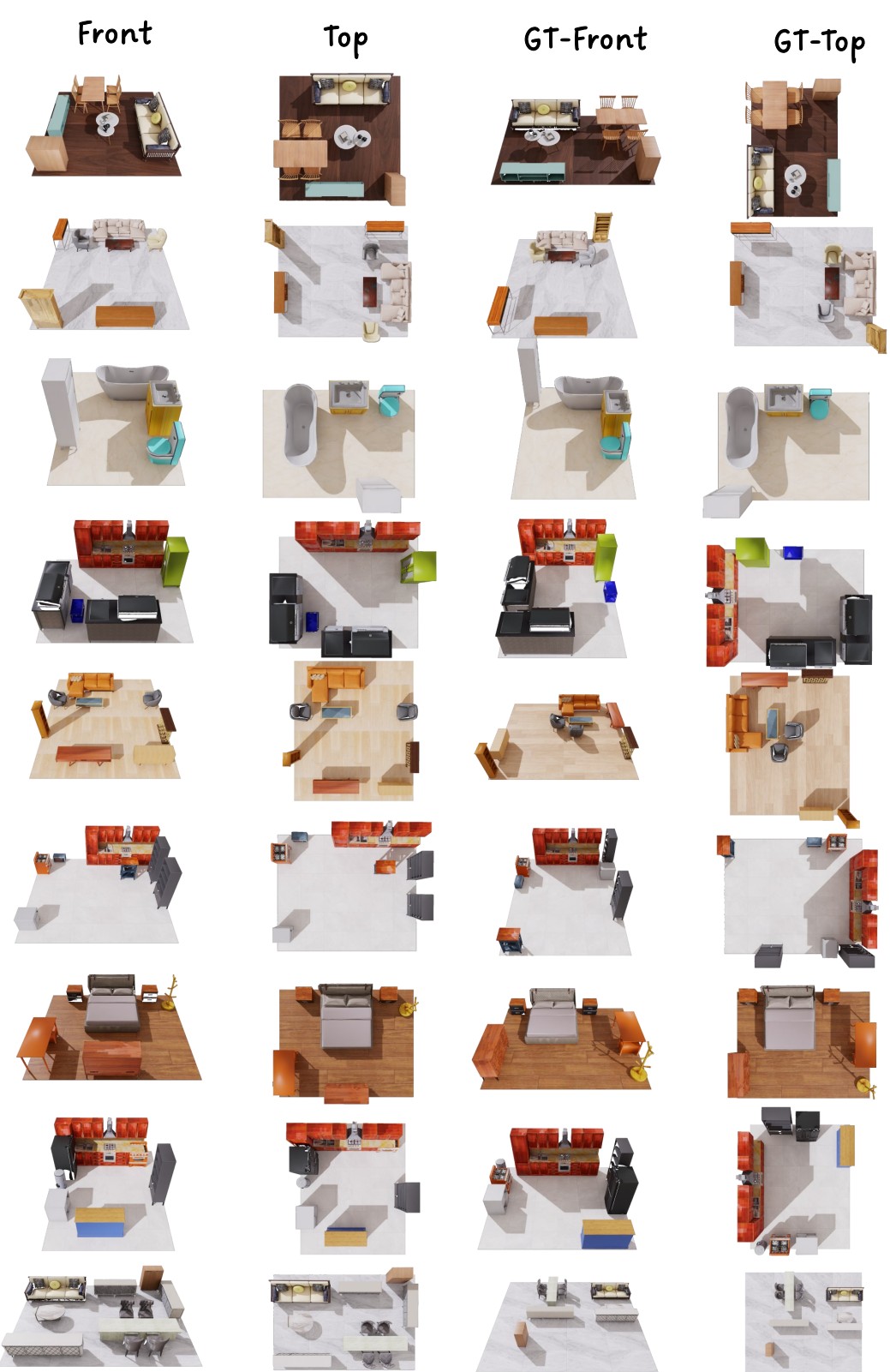

Figure 13: More qualitative results compared with the ground truth (Part 1).

Front  Top  GT-Front  GT-Top

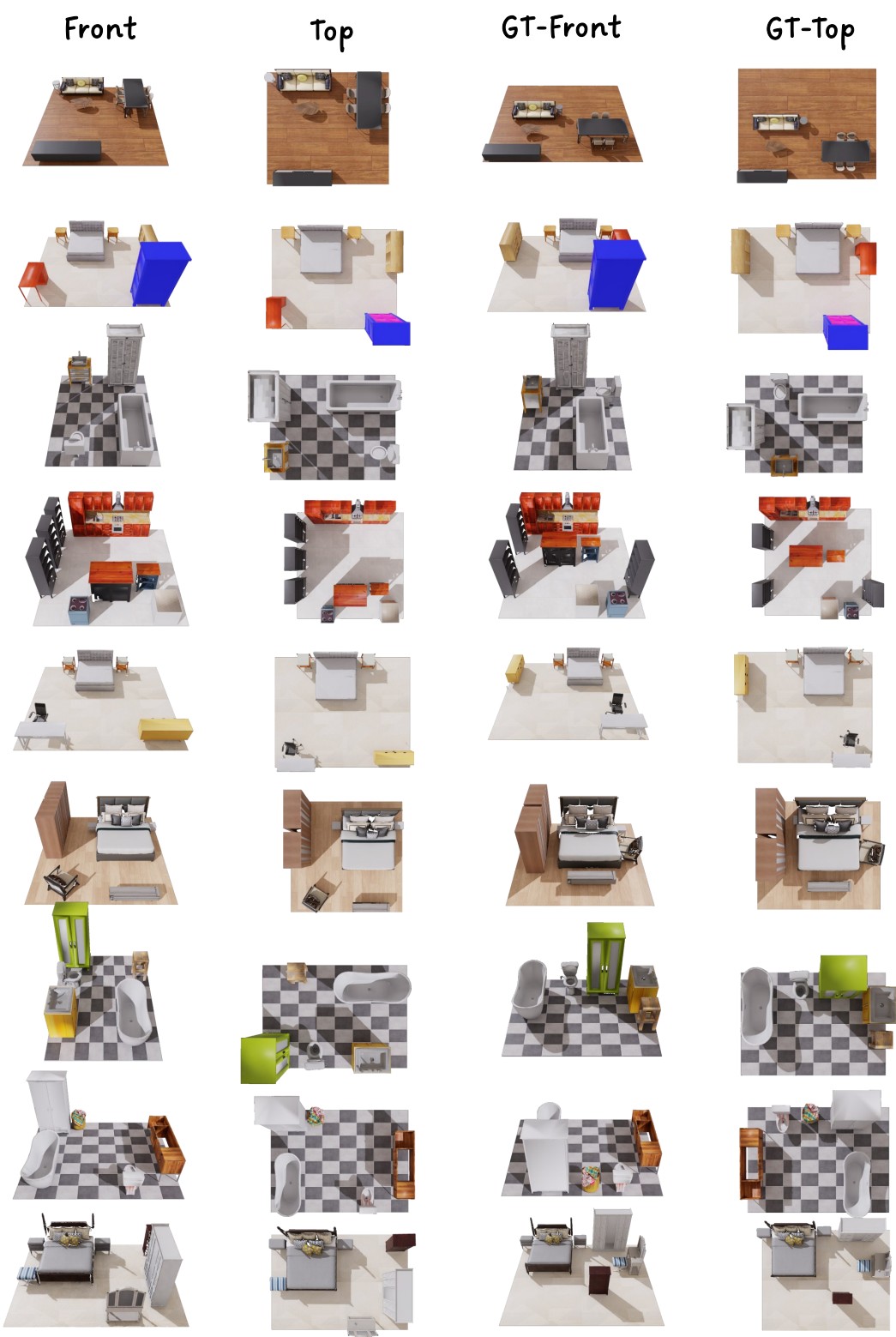

Figure 14: More qualitative results which are compared with the ground truth (Part 2).

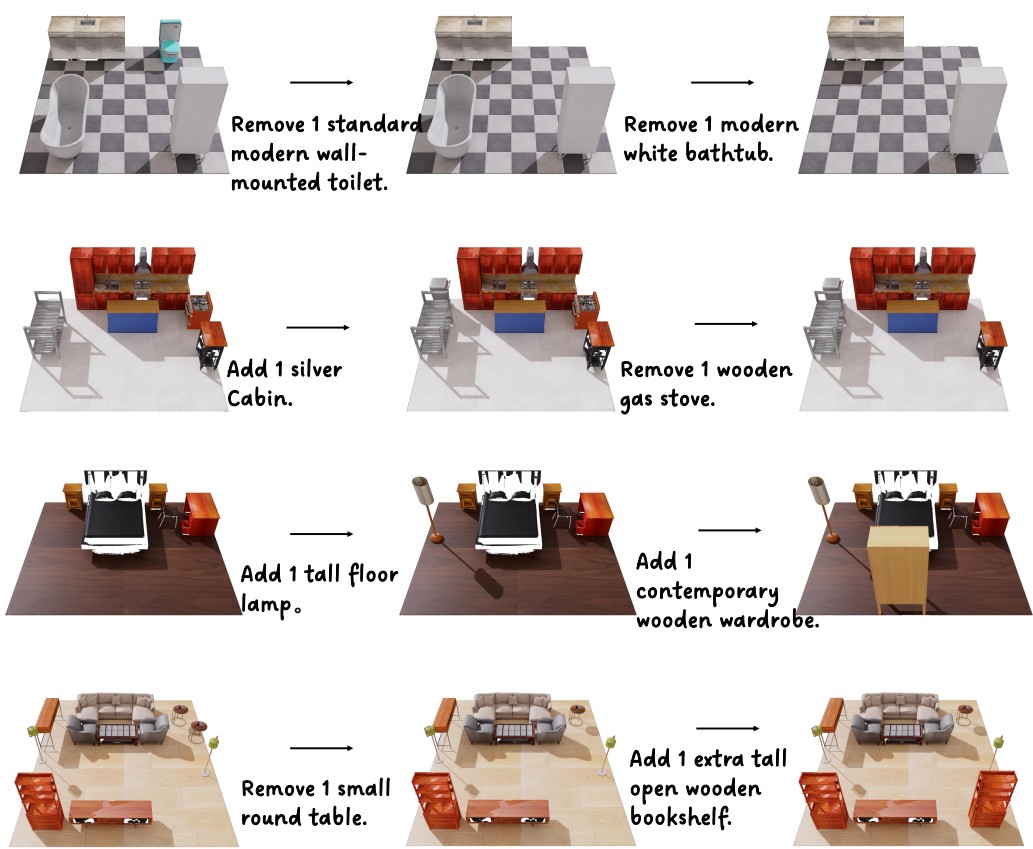

Figure 15: More editing results.

