# OpenReview forum: "OptiScene: LLM-driven Indoor Scene Layout Generation via Scaled Human-aligned Data Synthesis and Multi-Stage Preference Optimization"
_NeurIPS.cc/2025/Conference — NeurIPS 2025 poster_

### Official Review · Reviewer_1Sit · 2025-07-02

**Clarity:** 3
**Significance:** 3
**Originality:** 3
**Rating:** 4
**Confidence:** 4

**Summary:**

This paper proposes OptiScene, an indoor scene layout generation method based on the LLM. The authors start by constructing a large-scale indoor scene layout dataset, which contains about 17,000 scenes and covers four room types. Based on the dataset, they develop a two-stage training strategy, including supervised fine-tuning and multi-turn direct preference optimization, to produce structured layout representations from instructions. The results demonstrate the superiority of OptiScene over the prompt-driven and learning-based baselines. They also extend OptiScene to handle other relevant tasks, such as scene editing and robot navigation.

**Questions:**

1. Could the authors elaborate on the implementation details of the object retrieval module? Error accumulation maybe a key issue in current framework. That is to say, if the retrieval accuracy is not high, the final output will not conform to the user instruction.
2. Following up on the previous question, current evaluation metrics lack the assessment of the consistency with user instructions. How well can the output layout meet the user needs?
3. It Tab. 6, the introduction of semantic reasoning brings little performance improvement. Do the results mean that the semantic reasoning in SFT is not that necessary? Could the authors explain the results? It would be great if the authors could share the quantitative results of "SFT+DPO1+DPO2" without "Reasoning".
4. How is the layout editing task achieved using the layout generation model? Either the dataset construction and format is not clear.
5. The FID and GPT-4o Ratings are missing in the ablation studies.
6. In DPO-stage II, the authors add noise to GT layouts to produce negative samples. What is the noise level used? Do the authors study the effect of different noise levels? Are the negative samples produced by large noise better than those produced by small noise? It would be great if the author could have a discussion about this.
7. Does the current model support layout generation based on object relationships (e.g., the bed is to the east of the desk)? Users sometimes also specify the relative positions of the objects rather than just the types of the objects.

**Ethical Concerns:**

["NO or VERY MINOR ethics concerns only"]

**Final Justification:**

I am satisfied with the authors' rebuttal. In discussions with the authors, they also addressed my other concerns. I encourage the authors to include additional experiments and analyses in the final version. I will maintain my rating as borderline accept.

**Limitations:**

yes

**Quality:**

3

**Strengths And Weaknesses:**

Strengths:
- The curated large-scale dataset, 3D-SynthPlace, would be valuable for indoor layout generation.
- Two stage training of SFT along with DPO achieves layout generation effectively. Experimental results show that OptiScene is the SOTA method in this task.
- The introduction of the two-stage DPO framework, including human consensus alignment and geometric plausibility enhancement, makes the generated layouts closer to human preference and more geometrically valid.
- Paper writing is clear and easy to follow.

Weaknesses:
- Some notations and equations are confusing. For example, in Eq. 2, the LLM outputs $X$, which represents the input instruction. Besides, Does the so-called "bounding box" refer to the component size? It seems very easy to confuse with the concept of "layout" (i.e., $c$).
- The paper lacks some implementation details. For example, it is unclear how the object retrieval and layout editing (data construction, data format) are achieved.
- Introducing high-level semantic reasoning into the model output seems not effective.
- The ablation studies seem not thorough. Only two quantitative results are reported in this part.
- OptiScene generates layouts based on object types, but in real scenarios, users may also have requirements for the relative relationships between objects.

---

> ### Author Rebuttal · Authors · 2025-07-31
>
> We are grateful to Reviewer4 for the thoughtful assessment and positive evaluation of our work. We appreciate your recognition of 3D-SynthPlace, the two-stage DPO framework, and the clarity of our presentation.
> We now address your questions and concerns in detail.
>
> **R4Q1. How is the object retrieval module implemented, and how will it influence final scene quality?**
>
>   **Response:**
>
>   We follow a purely text-based retrieval approach, inspired by HOLODECK (Yang et al., 2024c), to match the objects mentioned in the user instruction to assets in our 3D object database:
>
> 1. Let $t'$ be the target object’s name from the wrapper, and $t$ be the annotation of a candidate 3D asset from the re-annotated the Objervase dataset.
>
> 2. We compute cosine similarity between their sentence embeddings using SBERT (`all-mpnet-base-v2`):
>
>    $$
>    T(t, t') = \cos\big(\mathrm{SBERT}(t),\,\mathrm{SBERT}(t')\big)
>    $$
>
> 3. The asset $o$ that yields the highest similarity is selected as the retrieved item for placement.
>
> This ensures semantic coherence even without explicit spatial constraints, making our retrieval robust against variations in descriptive text.
>
> **Retrieval Performance:**
>
> This retrieval approach has been validated within both Holodeck and LLPlace, and we have conducted dedicated tests on our retrieval module:
> - Explicit type queries (98% accuracy): When the input query explicitly specifies an object category (e.g., “sofa”, “desk”), the system returns the correct category as the top result 98% of the time.
> - Rich, free‑form descriptions:
>   - Top‑1 retrieval accuracy (86%): In 86% of cases, the single highest‑scoring asset exactly matches the ground‑truth asset.
>   - Top‑3 retrieval accuracy (94%): In 94% of cases, the ground‑truth asset appears somewhere among the three highest‑scoring candidates, providing a small ranked shortlist for follow‑up selection.
>
> Therefore, we believe the retrieval step has minimal impact on overall performance. Our 3D asset library is based on Objaverse, with refined and precisely annotated object labels, and is easily extensible beyond the 3D-SynthPlace dataset. The wrapper filters and standardizes user instructions, extracting clear room and object specifications. Retrieval then uses these parsed descriptions to match suitable assets. As long as the object type is clearly stated, retrieval remains reliable. For details on the wrapper, please see our response to **R3Q1(2)**.
>
> **R4Q2. How is consistency between the generated layout and user instructions evaluated?**
>
> **Response:**
>
> We fully understand your concern. In response to **R3Q1(2)**, we have provided a detailed explanation of how the Qwen2.5-7B wrapper is used to parse natural language instructions into a standardized JSON template, and extract information. This design helps eliminate redundant or irrelevant information and ensures a consistent input format.
> As further discussed in **R4Q1**, our text-to-asset retrieval strategy reliably maintains semantic alignment between the input and the generated layout.
> We also evaluated the model’s instruction-following ability under both **highly specific** and **vague prompts** (**see R3Q4**), assessing its robustness to varying instruction granularity and complexity.
>
> **R4Q3. Is semantic reasoning in SFT necessary, given the small performance gain shown in main paper's Table 6?**
>
> **Response:**
>
> Thank you for this insightful question. It is true that the performance gap between SFT with and without semantic reasoning appears small in terms of OOR. As shown in Table 6, the object overlap rates are nearly identical, and scene usability improves by only about 5 percentage points with reasoning.
> This is largely because, in the SFT-only setting, any object overlap renders the scene unusable. As a result, OOR dominates usability, and semantic reasoning has limited impact on the OOR metric itself.
> However, the reasoning step provides high-level spatial priors—descriptions of relative object positions—that guide the model toward more coherent and human-aligned layouts. While OOR may remain similar, this reasoning significantly improves overall scene structure and layout plausibility, leading to higher usability.
> To further clarify, we include below the quantitative comparison of SFT+DPO1+DPO2 with and without semantic reasoning:
>
> **Table 8:Compare the performance between with/without semantic reasoning**
> | SFT+DPO1+DPO2 | avg. OOR ↓ | UR (%) ↑ |
> |-|-|-|
> | with semantic reasoning | 0.023| 50|
> | without semantic reasoning  | 0.029| 42|
>
> These results show that although the OOR only slightly drops without reasoning, scene usability drops noticeably. This confirms that semantic reasoning helps the model produce more usable and human-preferred layouts, even if the overlap metric is not significantly affected.
>
> **R4Q4. How is the layout editing task implemented, and how is the dataset constructed and formatted?**
>
> **Response:**
>
> Thank you for your question. To enable layout editing using the same generation model, we constructed a dialogue-style editing dataset inspired by LLplace. Each original scene with more than four objects was converted into a two-turn dialogue:
> 1. In the first turn, we randomly removed a subset of objects (40% probability per object, excluding essential items like beds, tables, and sofas) to create a corrupted layout.
> 2. In the second turn, we either added or removed these objects depending on the task type.
> For the add task, we provided the corrupted scene and the list of removed objects as input (add=True), expecting the model to reconstruct the original full scene.
>  For the remove task, we gave the original scene and the target objects to remove (remove=True), expecting the model to generate the edited scene.
> Each sample includes structured object attributes (category, position, rotation) and uses <|eot_id|> to separate dialogue turns. In total, we created 5000 editing samples (3000 add, 2000 remove). The input format mirrors our original generation format, with added flags and object lists to indicate the editing intent.
> To adapt the model for editing, we further trained it for 5 additional epochs on this dataset, continuing with the original model. This improves the model’s ability to handle scene edits in a conversational context.
>
> **R4Q5. More ablation study results with GPT4o and FID.**
>
> **Response:**
>
> We provide a complete ablation study that includes FID and GPT‑4o scores to assess the contribution of each training component and the effect of high-level：
>
>   **Table 9: More Ablation Study with FID and GPT4o Score**
>   | Only SFT | Reasoning | SFT+DPO1| SFT+DPO1+DPO2 | OOR ↓ | UR ↑| FID ↓ | GPT4o ↑ |
> |-|-|-|-|-|-|-|-|
> | ✓ | - | - | - | 0.049 | 28%  |41.84|8.1|
> | ✓ | ✓ | - | - | 0.048 | 33%  |39.61|8.3|
> | ✓ | ✓ | ✓ | - | 0.033 | 40%  |37.24|8.3|
> | ✓ | - | ✓ | ✓ | 0.029 | 42% |35.51|8.6|
> | ✓ | ✓ | ✓ | ✓ | **0.023** | **50%** |**34.32**|**8.9**|
>
> As shown in the updated table, after incorporating both FID and GPT‑4o scores, the model's performance continues to improve progressively across different stages, exhibiting a clear stepwise enhancement. Each component contributes meaningfully to the generation quality,  validating the overall design of our pipeline.
>
> **R4Q6. What noise levels are used in DPO Stage II, and how do different levels affect performance?**
>
> **Response:**
>
> For this experiment, we tried three noise levels.  Our perturbations covered several types—objects placed outside the room, overlapping, and random placements.
>  Before settling on the current medium setting, we evaluated a heavy setting (severe overlaps, large out‑of‑room shifts, or fully random layouts) that made the preference gap between positive and negative samples too wide. We also tried a light setting, where objects were moved only slightly; this made the positive‑negative gap very small.
> We trained with each noise level and evaluated 100 scenes per setting with OOR and UR:
>
> **Table 10: Impact of Noise Levels on OOR and UR in Stage-II DPO**
> | Noise level | OOR ↓ | UR ↑  |
> |-|-|-|
> | Heavy       | 0.048 | 34% |
> | Medium | 0.023 | 50% |
> | Light       | 0.035 | 39% |
>
> While heavy noise leads to a sharp drop in OOR, it also results in the lowest usability. We believe this is because when negative samples are extremely poor, the model quickly learns to avoid only the most obvious errors. This causes it to converge prematurely, without learning to handle more nuanced spatial relationships—ultimately leading to layouts that still contain moderate overlaps or awkward placements.
> On the other hand, light noise provides minimal supervision beyond Stage I, offering little signal for further refinement. As a result, performance shows limited improvement and may even slightly degrade.
> The medium noise setting strikes the best balance: it introduces enough discrepancy to guide learning while avoiding extremes that cause saturation or instability. This produces the most effective training signal, yielding the best overall performance in Stage-II DPO.
>
> **R4Q7. Does the model support layout generation based on object spatial relationships?**
>
> **Response:**
>
> Thank you very much for this question. It also serves as a test of the model’s instruction-following capability. When processing user instructions through our wrapper, users may wish to specify object-to-object relationships as priors for the model input prompt. We have already discussed this issue in our responses to **R3Q4**, which we kindly refer you to for more details.
>
> **R4Q8: Some notations and equations are confusing.**
>
> **Response:**
>
> Thanks for the helpful suggestion. We acknowledge that some notations may be ambiguous or inconsistent. We will revise and standardize the notation in the final version to ensure clarity and avoid confusion with related concepts.

---

> ### Author Response · Authors · 2025-08-05
>
> Dear Reviewer,
>
> I hope this message finds you well. As the discussion period is nearing its end, I wanted to ensure we have addressed all your concerns satisfactorily. If there are any additional points or feedback you'd like us to consider, please let us know. Your insights are invaluable to us, and we’re eager to address any remaining issues to improve our work.
>
> Thank you for your time and effort in reviewing our paper.

---

> > ### Comment · Reviewer_1Sit · 2025-08-05
> >
> > Dear authors,
> >
> > Thanks for the substantial effort put into the rebuttal. The additional experiments and the corresponding analyses are highly appreciated. I carefully read through the response and found that it has addressed most of my previous concerns. To further strengthen the paper, I would encourage the authors to include some details in the revised version, including the implementation details of the object retrieval module, the dataset construction process for the editing task, and the additional quantitative results.
> >
> > I have a remaining point regarding the instruction-following evaluation. The test instructions, such as "Place the desk parallel to the bed" and "Place the chair next to the desk" seem to reflect strong dataset priors or biases. For instance, placing a chair next to a desk is a common pattern. As a result, the model might succeed in these cases without genuinely understanding or following the instruction, but simply by relying on learned co-occurrence patterns. This makes such examples less ideal for evaluating.
> >
> > Overall, I appreciate the authors' thoughtful response during the rebuttal.

---

> > > ### Author Response · Authors · 2025-08-06
> > > **Reviewer 1Sit Follow-Up Discussion (Part 1)**
> > >
> > > We sincerely thank the reviewer for the positive feedback and for recognizing the contributions of our work. We appreciate your thoughtful comments and will incorporate your suggestions in the final revision. Your insightful feedback will significantly strengthen our paper.
> > > Specifically, we will update the paper to include more details on the object retrieval module, the dataset construction process for the scene editing task, and additional quantitative results to further support our claims.
> > >
> > > And regarding the raised question, we also have additional analyses and experiments to provide further clarification and evidence, which we present below.
> > >
> > > **The clarification for the 'Chair next to the desk'**
> > >
> > > We sincerely thank the reviewer for raising this insightful concern. Indeed, the instruction "Place the chair next to the desk" is somewhat ambiguous, especially in rooms without a clearly defined orientation. In most training examples, the common placement pattern involves the chair being in front of the desk to allow users to sit and use the desktop.
> > > However, in our evaluation, we defined “next to the desk” as placing the chair on the **side** of the desk (i.e., along the shorter edge), rather than in front of it. The reported 80% success rate reflects this stricter definition. While chairs and desks are frequently paired in the dataset, side placement is much less common than front-facing configurations.
> > > Therefore, we believe this evaluation still demonstrates the model’s ability to follow spatial instructions to some extent, though we acknowledge it does not fully represent an out-of-distribution (OOD) case. To better assess instruction generalization, we designed a set of additional experiments, which are presented below.

---

> > > ### Author Response · Authors · 2025-08-06
> > > **Reviewer 1Sit Follow-Up Discussion (Part 2)**
> > >
> > > **More instruction following experiments**
> > >
> > > To further examine the model’s instruction-following ability, we design a set of targeted tests that go beyond simple patterns. These tests are in three categories: (1) out-of-distribution (OOD) object types, (2) explicit but uncommon spatial relations, and (3) semantically implausible object-room combinations.
> > > 1. OOD Object Type: Small Object Placement
> > > We evaluate the instruction: "Place a green mug on the desk." Since our training data contains no small objects, this setting tests the model’s ability to generalize spatial relationships to unseen categories. Among 20 generations：
> > > - 17 placed the mug successfully on the tabletop without overlap.
> > > - 3 showed varying degrees of collision or incorrect positioning.
> > >
> > > → Success Rate: 85% (17/20)
> > >
> > > This result demonstrates that the model can effectively generalize spatial placement behaviors to previously unseen object types, showing a basic understanding of relative positioning even without direct training supervision.
> > >
> > > 2. Explicit Spatial Relation: Wardrobe Next to Desk
> > > We evaluate the instruction: "Place the wardrobe next to the desk (on its side)." This object pair is not a common combination in our training data but expresses a clear spatial relation. Among 20 scenes:
> > > - 15 placed the wardrobe beside the desk as instructed.
> > > - 5 failed to establish the intended spatial relation.
> > >
> > > → Success Rate: 75% (15/20)
> > >
> > > These results indicate that even for uncommon or non-standard spatial relationships, the model can often generalize and fulfil user instructions correctly.
> > >
> > > 3. Room-Object Mismatch Instruction Test
> > > We further test whether the model can handle semantically implausible object-room combinations by inserting contextually inappropriate objects into each room type. For example:
> > > - Sofa in bedroom
> > > - Toilet in a living room
> > > - Desk in a bathroom
> > > - Bookshelf in a kitchen
> > >
> > > Each configuration is tested on 20 scenes, evaluated using three criteria: (i) whether the object is successfully placed, (ii) the Object Overlap Rate (OOR), and (iii) spatial usability (i.e., whether the resulting layout remains physically plausible and usable).
> > >
> > > **Table: Instruction-following test with contextually uncommon object insertions**
> > >
> > > |Room Type|Inserted Object|Add Success (%)|Overlap Rate|Spatial Usability (%)|
> > > |-|-|-|-|-|
> > > |Bedroom|Sofa|100|0.042|55|
> > > | Living Room |Toilet|100| 0.078|35|
> > > |Bathroom|Desk|100|0.061| 40|
> > > |Kitchen|Bookshelf|100| 0.065| 40|
> > >
> > > Thanks to our robust wrapper, the model is able to successfully add the requested objects into the scene, even when the instructions are semantically implausible. As shown in Table, the placement success rate remains 100% across all cases.
> > > However, we observe a clear increase in OOR compared to normal scene generation. While these values remain within an acceptable range, they reflect the added spatial difficulty when incorporating contextually inappropriate objects.
> > > To further evaluate the effect of such OOD insertions, we also consider spatial usability. Although usability scores drop in these abnormal cases, the model often leverages its reasoning and generalization ability to place objects in relatively plausible positions. For example, sofas and toilets are typically aligned along walls, while desks and bookshelves tend to appear near sinks or storage units in bathrooms and kitchens, respectively. We would like provide figures of these scenes in the future revision paper.
> > >
> > > These three experimental settings collectively demonstrate the model’s ability to (1) follow instructions involving unseen object types, (2) interpret rare object relationships, and (3) handle room-object mismatched instructions, all of which extend beyond simple co-occurrence priors in the training data.
> > >
> > > We thank the reviewer for the constructive feedback. We remain open to addressing any further concerns or clarifications that may arise.

---

> > > ### Author Response · Authors · 2025-08-09
> > > **Follow-up on our rebuttal responses for Reviewer 1Sit**
> > >
> > > Dear Reviewers 1Sit,
> > >
> > > We sincerely thank you for your professional and valuable reviews. Your insightful feedback will greatly help strengthen our paper.
> > >
> > > We have carefully addressed your latest questions. As the deadline is approaching, if you have any further concerns, please let us know, and we will do our best to respond promptly.
> > >
> > > We would also like to gently remind you that it seems the **Mandatory Acknowledgement** has not yet been completed. We would greatly appreciate it if you could kindly confirm the acknowledgement at your earliest convenience.
> > >
> > > Thank you for your time and effort in reviewing our paper.

---

### Official Review · Reviewer_xtFJ · 2025-07-02

**Clarity:** 2
**Significance:** 3
**Originality:** 2
**Rating:** 4
**Confidence:** 5

**Summary:**

The paper introduces OptiScene, an LLM-based framework for preference-aware indoor layout generation. Unlike prior prompt-driven approaches that struggle with controllability and physical understanding, and learning-based approaches that suffer from scarce data sources leading to a lack of expressiveness, OptiScene addresses these concerns by post-tuning an LLM for the spatial design task using the proposed large-scale 3D-SynthPlace dataset.

This is achieved in two steps: The SFT stage introduces a high-level semantic reasoning step before layout generation, while the two-stage DPO pipeline helps align layout generation with human preferences and physical constraints.

**Questions:**

All of my questions are listed in the weaknesses section, and I may adjust the rating if they are well addressed. Specifically:

1) Some parts of the method, particularly the SFT process, require further clarification.
2) Additional experiments could be conducted to evaluate the method’s behavior in challenging cases (e.g., valid overlapping bounding boxes) and with complex, ambiguous free-form instructions reflecting real-world scenarios.
3) Potential bias in the training data should be discussed, and generalization experiments should be conducted.

**Ethical Concerns:**

["NO or VERY MINOR ethics concerns only"]

**Final Justification:**

The authors have addressed my questions and provided further clarifications, which resolved most of my concerns. I am inclined to accept this paper. I have recommended that the authors include these discussions and experiments in the final version to strengthen the work.

**Limitations:**

Yes

**Quality:**

2

**Strengths And Weaknesses:**

Strengths:

1)	The proposed pipeline attempts to combine the strengths of prompt-driven and learning-based methods, and further extends them by aligning the generation process with human preferences.
2)	The paper introduces a synthetic indoor scene dataset called 3D-SynthPlace, which covers a wide variety of layouts along with semantic summaries and scene descriptions, contributing to the field.
3)	The proposed approach demonstrates potential for different applications such as scene editing and object-centric navigation.

Weaknesses:

1)	Missing details for SFT: Although Section 3.3. discusses the SFT process, some crucial details are missing from the text or require further clarification:  a) L195-196: How is the supervision provided to minimize the discrepancy between predictions and ground-truth layouts, including both the semantic summary and the structured output? b) How is the user instruction U converted into the JSON specification X? c) Generally, object retrieval from the dataset is performed using the Euclidean distance between the predicted bounding boxes and the bounding boxes of the dataset assets, as well as category information. However, Equation 1 appears to use object descriptions for retrieval. How this is achieved is not clear.
2)	Valid overlapping bounding boxes: DPO Stage 2 and the way prompts are constructed (L196–197) are designed to avoid object collisions. However, real-world scenarios often include cases where valid overlapping bounding boxes naturally exist (e.g., a chair underneath a desk). How does the method handle these cases? Is it possible to include them as challenging positive examples?
3)	Possible bias in the training data: a) Despite the advanced filtering, the training dataset relies on Holodeck’s predictions and may inherit its biases. b) During DPO training, negative samples are generated by the SFT model itself, which could lead to similar error patterns and limited diversity.
4)	Lack of ablations on user instruction complexity: No evaluation is performed showing how the free-form instructions which may exhibit high complexity and ambiguity affect the generations.

---

> ### Author Rebuttal · Authors · 2025-07-31
>
> We sincerely thank you for the encouraging feedback and thoughtful summary. We are pleased that you found our dataset, two-stage DPO framework, and overall method design valuable.
> Below, we address your comments and respond to the specific concerns you raised.
>
> **R3Q1. Clarify key missing details in the SFT stage: supervision methods, instruction-to-JSON conversion, and the object retrieval process?**
>
> **Response:**
>
> We agree these aspects were not sufficiently clear in our manuscript. We clarify each sub-question explicitly below, and will incorporate these detailed explanations into our revised version:
>
>  1) Supervision during SFT:
>
> During SFT, the model jointly generates (A) a semantic summary $S^{\text{sem}}$ and (B) a structured JSON layout with 3D coordinates $c_{1\ldots N}$ and orientations $r_{1\ldots N}$. We serialize both outputs into token sequences and apply token-level cross-entropy loss against human-annotated ground truth. This unified supervision encourages the model to capture both global semantic intent and precise spatial configurations in a single forward pass.
>
> 2) Instruction-to-JSON Wrapper:
>
> We use a Qwen2.5-7B-Instruct LLM as a wrapper to convert natural language user instructions $\mathcal{U}$ into structured JSON $\mathcal{X}$. The process works as follows:
> - The wrapper extracts **room size**, **room type**, **object types**, and **object counts** from the instruction.
> - If any field is missing, the wrapper heuristically infers it based on existing information (e.g., number and type of objects).
> - It then retrieves suitable 3D assets for each object (details in part 3).
> - The extracted values are inserted into a predefined JSON layout template, which is then used as part of the prompt input to OptiScene for inference.
> This wrapper provides robustness by normalizing diverse instructions, modularity for easier debugging, and flexibility through noise filtering. It is also easily extensible—indeed, we further extended it in R3Q4 to extract spatial relationships between objects.
>
> 3) Object Retrieval Process:
>
> We follow a purely text-based retrieval approach, inspired by HOLODECK (Yang et al., 2024c), to match the objects mentioned in the user instruction to assets in our 3D object database.
> Due to the character limitation, please refer to the **R4Q1** for more details, where we discuss the object retrieval process and show the performance test of the text-based approach.
>
> **R3Q2. How does the method handle naturally valid bounding-box overlaps? Are these explicitly modelled?**
>
> **Response:**
>
> Certain overlaps (e.g., chairs under desks) naturally occur in realistic scenes, and explicitly handle these cases as follows:
> - Dataset Preservation: We intentionally retain the valid overlapping layouts (e.g., chairs tucked beneath desks) in our synthetic dataset—474 scenes (2.8% of total), predominantly in living rooms—allowing the model to recognize natural overlap distributions without skewing its collision avoidance capability.
> - Prompt Engineering (Appendix Table 7): Our prompts explicitly instruct minimal distances for specific object pairs (e.g., chair–desk), guiding the model to correctly interpret these overlaps as intended and acceptable rather than as collisions.
> - Stage 2 DPO Noise Injection: During DPO Stage 2, we preserve chair–desk overlaps as positive examples, avoiding random perturbations on these pairs. Conversely, overlaps involving inappropriate object pairs (e.g., sofa–sofa intersections) remain labelled as negative examples.
> Thus, our approach robustly differentiates between valid overlaps and unintended collisions, ensuring realistic, high-quality layout generation.
>
> **R3Q3. How does the method handle biases inherited from Holodeck and limited negative sample diversity during DPO training?**
>
> **Response:**
>
> We clarify that our training dataset is **not directly derived from raw Holodeck outputs**. Instead, we carefully filtered and applied handcrafted refinements to ensure data quality and alignment with human preferences. Nonetheless, we acknowledge the possibility of **residual biases** from the Holodeck remaining in our data, which we addressed systematically:
> - Manual Selection of Positive Samples: During the DPO stage 1, we manually selected layouts that closely align with human preferences, ensuring positive samples specifically counteract residual biases inherited from Holodeck and original 3D-FRONT datasets.
> - Two-stage DPO Training: To enhance the diversity and effectiveness of negative samples, we introduced a DPO Stage 2. We deliberately applied controlled perturbations—including slight random displacements, overlaps, and out-of-bound placements—to relatively high-quality layouts, increasing the model’s ability to robustly differentiate valid and invalid configurations.
> - Temperature-based Diversity for Negative Samples:  Additionally, during DPO Stage 1, we generated negative samples by varying the sampling temperature from the SFT model. Consistently similar negative outputs at different temperatures indicate persistent errors that require repeated exposure, whereas varying outputs suggest multiple error patterns beneficial for robust training. Moreover, using SFT-generated negative samples for DPO training is a validated approach, as demonstrated in prior works [1,2].
>
> [1] Enhancing LLM Reasoning with Iterative DPO: A Comprehensive Empirical Investigation
>
> [2] Iterative reasoning preference optimization
>
> **R3Q4. How does instruction complexity affect generation performance?**
>
> **Response:**
>
> To support consistent layout generation across varying instruction types, we first enhance the wrapper, which parses natural language instructions into structured JSON representations by extracting room attributes, object descriptions, and (when available) object-to-object spatial relationships.
> We design the wrapper to handle two main cases:
> - For highly specific and directional prompts, we use the wrapper to extract spatial relations and then encode them as **[objects relationship]** tokens, which are injected at inference time to guide generation without requiring retraining.
> - For vague or underspecified prompts, where explicit object information is lacking, we rely on the wrapper to creatively infer plausible object categories and quantities from contextual cues.
>
> **Instruction Complexity Evaluation**
>
> We first assessed the model’s robustness to instruction complexity by categorizing prompts into three levels: Simple (≤5 objects, precise object description and number), Medium (6–15 objects, moderately vague description with some object relationships), and Complex (15-20 objects, highly ambiguous).
> Each group contains 50 prompts evaluated on living room scenes. The wrapper and retrieval process achieves over 95% parsing success across all levels. However, under complex or vague instructions such as “some chairs” or “a few shelves,” the number of extracted objects tends to vary (typically between 2–4), leading to decreased layout reliability.
>
> **Table 5. Generation performance under different instruction complexity levels**
> | Instruction Level |  avg. OOR ↓ | avg.GPT-4o ↑ | UR (%) ↑ |
> |-|-|-|-|
> |Simple|0.021|8.9|70|
> |Medium| 0.087 |7.9|45 |
> |Complex| 0.186 |7.1| 30 |
>
> Despite the increased difficulty, our model maintains reasonable performance even under complex instructions, achieving an average GPT-4o score of 7.1 and a usability rate of 30%. This demonstrates the robustness and stability of both our wrapper and the generation model when handling ambiguous or underspecified inputs.
>
> **Instruction-Following with Extreme Prompts**
>
> We further evaluate how the model handles two extreme instruction types:
>  (1) Highly specific and directional instructions,
>  (2) Extremely vague and underspecified instructions.
>  Each of these settings was evaluated over 20 test cases across four room categories.
>
> **Directional Instructions**
>
> To evaluate the model's ability to follow directional prompts, we measure both scene-level and instruction-level relation success rates, along with OOR and scene usability, as summarized in Table 6.
> A scene is considered successful only if all specified spatial relations are satisfied. Under this criterion:
> - 12/20 scenes met all spatial constraints (60% scene-level success)
> - 30/46 individual relations were followed (65.2% instruction-level success)
>
> **Table 6: Performance on directional prompts**
>    | Metric | Value  |
>    |-|-|
>    | Scene-level Relation Success Rate|60%|
>    | Instruction-level Relation Success Rate | 65.2% |
>    | OOR |  0.056 |
>    | Scene Usability |  40% |
>
> We also evaluated single-instruction examples. For instance:
> - “Place the desk parallel to the bed” → correct placement in 66.7% of 12 cases
> - “Place the chair next to the desk” → correct placement in 80% of 20 cases
>
> These results suggest that, despite not being explicitly trained on spatial language, the model exhibits a limited but notable understanding of spatial semantics. However, the model often fails with instructions involving corner placements, likely due to the absence of explicit geometric encoding of room corners.
>
>
> **Vague Prompts**
>
> To assess robustness under ambiguous inputs, we tested prompts containing vague quantifiers (e.g., “some chairs”) or high-level intents (e.g., “a cozy living space”). We report JSON parse success and scene usability:
>
> **Table 7:Performance on vague instructions**
>    | Metric | Value |
>    |-|-|
>    |JSON Parse Success Rate|65%|
>    |Scene Usability|35%|
>
> In 65% of cases, the wrapper generated valid object layouts, but many still suffered from usability issues (e.g., missing object types or implausible combinations).
>  The remaining failed parsing due to overly abstract input or absence of essential categories (e.g., no sofas in a living room).
> While the wrapper handles moderately vague prompts reasonably well, it struggles with highly underspecified instructions. We plan to explore this direction in future work.

---

> ### Author Response · Authors · 2025-08-05
>
> Dear Reviewer,
>
> I hope this message finds you well. As the discussion period is nearing its end, I wanted to ensure we have addressed all your concerns satisfactorily. If there are any additional points or feedback you'd like us to consider, please let us know. Your insights are invaluable to us, and we’re eager to address any remaining issues to improve our work.
>
> Thank you for your time and effort in reviewing our paper.

---

> > ### Comment · Reviewer_xtFJ · 2025-08-06
> > **Official Comment by Reviewer xtFJ**
> >
> > Dear authors,
> >
> > Thank you for addressing my questions and providing further clarifications. Your rebuttal resolves most of my concerns, and I will be increasing my score accordingly. I strongly recommend including the details regarding the SFT stage, the discussion on valid overlapping bounding boxes, and the experiments on instruction complexity in the final version. These additions would significantly strengthen the paper and provide a more comprehensive understanding for readers. I appreciate all the effort you have put into the rebuttal!

---

> > > ### Author Response · Authors · 2025-08-06
> > >
> > > We sincerely thank the reviewer for the encouraging feedback and for the decision to increase the score. We truly appreciate your recognition of our efforts in the rebuttal.
> > >
> > > We will incorporate all the suggested additions into the final version, including (1)A detailed description of the SFT stage; (2) A clearer discussion on naturally valid overlapping bounding boxes; (3) And more experiments on instruction complexity and their implications.
> > >
> > > We believe these revisions will indeed make the paper more complete and accessible to readers, and we are grateful for your constructive guidance!

---

### Official Review · Reviewer_LV9F · 2025-07-03

**Clarity:** 3
**Significance:** 2
**Originality:** 2
**Rating:** 4
**Confidence:** 4

**Summary:**

This paper aims to the problem of scene layout generation from a natural-language description of a set of indoor objects. It introduces **OptiScene**, an LLM‑based framework fine-tuned through SFT and RL (DPO [22]). Additionally, the manuscript also introduces **3D-SynthPlace** a novel dataset constructed from Holodeck[35], significantly extending 3D-Front[9] dataset.

**Questions:**

* Q1: in L302, Object-Centric navigation experiments are presented. The experiment description defines a command "find a <object> in the room" to execute a robot exploration. Results are presented in Tab 5. Can the authors elaborate on how the robot/agent explores the scene to find the object? If the robot does not use the proposed **OptiScene** why this experiment is evidence that the proposed solution allows the downstream task of navigation?

* Q2: The ablation study in Tab. 6 indicates that prompting with a high-level description (reasoning) significantly boosts performance (as pointed out in L314). I’m curious whether other formats of high-level input, such as a JSON file with some centroids or even visual inputs (image in BEV), would have a similar effect or not. How might the results change with these different formats?

* Q3: How the proposed solution performance with more complex scene layouts? For instance 5, 10, 20, 30 objects. Is there any limitation in this matter.

**Ethical Concerns:**

["NO or VERY MINOR ethics concerns only"]

**Final Justification:**

I partially agree with the authors about OptiScene’s role in the navigation task as presented in the current manuscript. From my understanding, using OptiScene increases the amount of training data available for the navigation tasks (as the authors mentioned in their rebuttal, ' it helps expand the data space for training and testing'). However, the experiments showed in the manuscript and in the rebuttal do not convey this idea. OptiScene was mainly used for testing predefined solutions for navigation. The impact of OptiScene for training was not presented. Additionally, it remains unclear to me how OptiScene may helps for navigation task in larger environments, with multi-rooms, corridors, and multiple areas, which is the main setting for the embodied navigation tasks.

Nevertheless, based on the authors' response R2Q3, I want to emphasize the clear contribution of this paper. Although, the proposed solution’s performance declines as the number of objects increases (i.e., which is totally expected), it still significantly outperforms the baselines. The selected baselines are both appropriate and fair, and the current results provide strong evidence for the manuscript’s contribution.

Even though, there are some minor flaws, I keep my rating as borderline accept.

**Limitations:**

yes

**Paper Formatting Concerns:**

No format issues

**Quality:**

3

**Strengths And Weaknesses:**

# Strengths
* The paper is well-written and explain clearly the motivation behind the proposed solution. It offers a clear view of the current modules and contributions, and effectively evaluates the proposed method/solution in a suitable benchmark with valid baselines for scene layout generation.

# Weaknesses
* The current manuscript fails at explaining and discuss the limitation of the current solution. For instance, more complex scenes (specifically more objects) and un-seen items (objects/items no presented during training).

---

> ### Author Rebuttal · Authors · 2025-07-31
>
> We thank Reviewer 2 for the clear summary and positive evaluation. We appreciate your recognition of our motivation, modular design.
> Below, we address your concerns and provide clarifications on the specific points you raised.
>
> **R2Q1. Clarification on Object-Centric Navigation (L302)**
>
>   **Response**:
>
> Here is our answer about navigation with LLM:
>
> We first provide clarification on the navigation setting as follows:
>
>   For navigation experiments, we adopt the settings in Benchmark 1: Object Loco-Navigation in InfiniteWorld [1]. Specifically, the process by which a robot/agent explores a scene to searches for a target object is as follows: First, a 2D grid representation of the environment is constructed through an occupancy map, which divides the space into free zones, obstacle zones, and unknown zones. This grid serves as the foundational spatial reference for subsequent path planning, enabling the agent to identify navigable areas and avoid obstacles. Then, natural language instructions such as "Find an [object] in [room]" are parsed: LLM extracts semantic information from the instruction to determine the target object (e.g., "black bedside table") and its approximate location clue (e.g., "in the bedroom"), and further queries the scene's built-in object coordinate data or real-time scene graph to obtain the specific coordinates of the target object. Subsequently, the path follower, based on the 2D grid, uses an algorithm to plan the optimal path from the agent's current position to the target coordinates. If the target coordinates fall within an obstacle zone, the nearest non-colliding point is selected as an alternative target. During movement, the agent proceeds along the planned path, with the final requirement that the target object is within its 60-degree horizontal field of view and within 2 meters. If these field-of-view conditions are not met due to occlusion or other reasons, the task fails.
>
>   Then we discuss the connection from navigation to OptiScene:
>
>  Although the navigation algorithm itself is not modified, we evaluate it on scenes generated by OptiScene. Since indoor datasets suitable for such embodied tasks are limited, our method helps expand the data space for training and testing. The fact that existing navigation algorithms can be directly applied to scenes generated by OptiScene, and achieve comparable behavior, validates that our synthetic scenes are well-aligned with real ones and usable for downstream embodied tasks such as navigation.  And also, we think navigation is one kind of way to evaluate the usability of the generated indoor scenes. As in **R1Q3**, we introduce navigability to test models' performance. Although navigability is different from the object loco-navigation results, they both show the overall spatial rationality of the scene.
>
>   Ref:
>
>   [1] Ren, P., Li, M., Luo, Z., Song, X., Chen, Z., Liufu, W., ... & Liang, X. (2024). Infiniteworld: A unified scalable simulation framework for general visual-language robot interaction
>
> **R2Q2. Is the semantic reasoning step part of the input prompt or the model’s internal reasoning process? Would alternative high-level inputs (e.g., fixed positions, centroids, BEV images) also improve performance?**
>
> **Response:**
>
> Thanks for your question! For the first part of the question, we apply the semantic reasoning because we believe that if we let the model try to understand the relationship between each other can help to generate more reliable 3D scenes. So we construct this data in the training dataset and **make the semantic reasoning as a step in the chain of thought before outputting the final 3D scene result**. So this is not a part of the prompt for the model; we just try to let the model learn object-object relationships of the scenes.
>
>  And for the second part of the question. As we understand, the reviewer would like us to test fixing the positions of certain objects as the input. For example, when generating a bedroom, first specifying the bed’s position and then predict the positions of the remaining objects. We believe this experiment is interesting and clearly demonstrates our model’s robustness, so we performed inference on four room categories while fixing the positions of some selected objects (e.g., Bed, sofa, bathtub...). This setup also aligns closely with the downstream task of scene editing we discussed in the paper. In addition to directly performing inference with fixed object positions, we further included a fine-tuned version of our model specifically trained for editing as a baseline comparison.  And we provide the quantitative results based on key metrics, Object Overlap Rate (OOR), GPT4o, and Usability:
>
> **Table 3: Compare the Fixed Position OptiScene with other settings**
> | Condition| avg. OOR ↓| avg. GPT4o↓ | Scene Usability (%) ↑ |
> |-|-|-|-|
> | Original OptiScene | 0.023 | 8.7| 50|
> | Editing OptiScene  | 0.028 | 8.7| 50|
> | Fixed Position OptiScene | 0.045 | 8.2| 40|
>
> As shown in Table 3, even when fixing the positions of certain objects during inference, the model maintains good performance with only a slight drop, which demonstrates our model's robustness.
>
> Additionally, we conducted further experiments using highly specific user instructions that include object relationship descriptions (see **R3Q4** for details). The results show that, without any re-training, our model can leverage the generalization ability of the LLM to interpret such structured instructions and still generate reliable 3D scenes.
> But our model is designed to use an LLM to generate spatial placements of indoor objects and does not currently integrate other modalities (like 2D image or 3D point cloud) or change the overall architecture to help the scene generation. We agree that introducing other modalities is an exciting direction, and we plan to explore it in future work.
>
>
> **R2Q3. How do the proposed solutions perform with more complex scene layouts? For instance, 5, 10, 20, 30 objects. Is there any limitation in this matter?**
>
> **Response:**
>
> Thank you for your insightful question. Evaluating the robustness of our model under varying scene complexity is indeed crucial. To address this, we conducted comprehensive experiments on scenes containing 5, 10, 15, 20, and 30 objects, sampling 20 living-room scenes for each scenario and evaluating performance using Object Overlap Rate (OOR), FID scores, GPT-4o ratings, and usability metrics. We compared our results against several baselines, including DiffuScene, LLPlace, and Holodeck. While Holodeck's OOR always be 0 due to its processing step that enforces, we won't compare Holodeck's OOR in the table.
> As summarized in the table below, our method consistently outperforms baselines (DiffuScene, LLPlace, Holodeck) across most complexity levels, maintaining strong performance up to 15 objects. However, at higher complexities (20+ objects), especially at 30 objects, the model exhibits significant performance drops (e.g., OOR rises to 0.289, usability drops to 0%), clearly indicating a limitation.
> We attribute this decline primarily to the scarcity of densely-packed scenes in our training data, resulting in an out-of-domain scenario at very high object counts. This limitation is prevalent across other state-of-the-art methods as well as Holodeck, for example, completely fails to generate coherent layouts at 30 objects, even after we change the code to enlarge room sizes to accommodate them.
>
> **Table4. Performance Across Object Counts (avg. OOR ↓ / avg. GPT‑4o ↑ / UR ↑)**
>
> |Method|5 Objects|10 Objects|15 Objects|20 Objects|30 Objects|
> |-|-|-|-|-|-|
> |**Ours**| **0.020 / 9.3 / 75** | **0.047 / 8.7 / 60** | **0.088 / 8.1 / 35** | **0.154 / 7.0 / 20** | **0.289 / 6.8 / 0** |
> |DiffuScene|0.075 / 8.3 / 45 | 0.082 / 8.0 / 45  | 0.093 / 7.6 / 30  | 0.211 / 6.5 / 10 | 0.348 / 6.1 / 0  |
> |LLplace|0.072/8.5/40| 0.085 / 7.8 / 35  | 0.129 / 7.6 / 25  | 0.186 / 6.3 / 5 | 0.292 / 5.5 / 0  |
> |Holodeck|- /8.9/65| - /8.5/60|- /7.8/30| - /7.3 / 25|- / 6.9 / 0|
>
>
> As for the unseen objects question. We discuss the concern about how our model handles previously unseen object descriptions as follows:
> - (1) Retrieval-based Generalization:
>  As discussed in **R3Q4** and **R4Q1**, our model relies on a retrieval module that maps natural language object descriptions to corresponding 3D assets. This module achieves a 98% retrieval accuracy for matching object classes, ensuring reliable selection even for unseen or varied descriptions.
> - (2) LLM Generalization + Test-time Coverage:
>  The LLM demonstrates strong generalization ability in interpreting semantically novel or rare descriptions (e.g., mapping “seats” to “chair”).
>  Moreover, our test set already contains many object descriptions not seen during training, and the model successfully handles them, confirming its robustness to unseen inputs.
> - (3) Focus on Large Floor-based Furniture:
>  Our task targets large furniture items that are placed directly on the floor (e.g., beds, sofas, wardrobes), rather than small or decorative objects. This reduces ambiguity and improves consistency in object grounding.
> - (4) High-quality 3D Asset Pool:
>  All retrieved assets come from our annotated 3D database based on Objaverse, ensuring sufficient diversity and realism to support generalization.
>
> In summary, our model demonstrates robustness and quality in moderately complex scenes, yet encounters significant challenges at extremely dense object arrangements with an extremely large number of objects (>20).
> To improve performance in such scenarios, expanding the training set to include richer and denser configurations, combined with targeted data augmentation strategies, will be essential future directions.
> Moreover, we believe our model is capable of handling unseen objects during inference in the context of indoor scene layout generation.

---

> ### Author Response · Authors · 2025-08-05
>
> Dear Reviewer,
>
> I hope this message finds you well. As the discussion period is nearing its end, I wanted to ensure we have addressed all your concerns satisfactorily. If there are any additional points or feedback you'd like us to consider, please let us know. Your insights are invaluable to us, and we’re eager to address any remaining issues to improve our work.
>
> Thank you for your time and effort in reviewing our paper.

---

> ### Comment · Reviewer_LV9F · 2025-08-05
>
> I partially agree with the authors about OptiScene’s role in the navigation task as presented in the current manuscript. From my understanding, using OptiScene increases the amount of training data available for the navigation tasks (as the authors mentioned in their rebuttal, *' it helps expand the data space for training and testing'*). However, the experiments showed in the manuscript and in the rebuttal do not convey this idea. OptiScene was mainly used for testing predefined solutions for navigation. The impact of OptiScene for training was not presented.  Additionally, it remains unclear to me how *OptiScene* may helps for navigation task in larger environments, with multi-rooms, corridors, and multiple areas, which is the main setting for the embodied navigation tasks.
>
> Nevertheless, based on the authors' response R2Q3, I want to emphasize the clear contribution of this paper. Although, the proposed solution’s performance declines as the number of objects increases (i.e., which is totally expected), it still significantly outperforms the baselines. The selected baselines are both appropriate and fair, and the current results provide strong evidence for the manuscript’s contribution.
>
> Even though, there are some minor flaws, I keep my rating as **borderline accept**.

---

> > ### Author Response · Authors · 2025-08-06
> > **Reviewer LV9F Follow-Up Discussion**
> >
> > We sincerely thank the reviewer for the detailed and balanced feedback. We are also grateful for your recognition of our rebuttal efforts. Your insightful comments will greatly help us strengthen the final version of the paper.
> >
> > And also, we would like to take this opportunity to clarify the intended role and interpretation of the navigation task in our work.
> >
> > Our primary objective is to generate semantically meaningful and geometrically plausible furniture layouts for indoor scenes. As the reviewer correctly noted, OptiScene achieves consistent improvements over strong baselines in this regard.
> > **And the navigation task is not the central goal of our method, but rather serves as a proxy evaluation to assess whether the generated layouts are usable in downstream embodied tasks.** Since well-established navigation algorithms are sensitive to layout quality, their success rates can indirectly reflect the realism and spatial coherence of the generated scenes.
> >
> > We acknowledge, however, that our original manuscript did not fully demonstrate this point. Specifically, Table 5 in the main paper only evaluated OptiScene layouts, without comparing them to those generated by other methods. To address this, we conducted additional experiments on the object loco-navigation task, comparing OptiScene against DiffuScene and LLplace using a fixed navigation policy across all methods. The results are shown below:
> >
> > **Table : Compare the OptiScene with other methods on the object loco-navigation task**
> > | Layout Method | LLM               | Success Rate | Navigation Error  |
> > |-|-|-|-|
> > | OptiScene | GPT-4o            | 87.56%       | 1.00             |
> > |               | Qwen-3-8B         | 75.92%           | 0.98                |
> > |               | ChatGLM4-Flash    | 72.33%           | 0.92                |
> >   | LLplace   | GPT-4o            | 73.12%           |         1.00         |
> > |               | Qwen-3-8B         | 64.50%           |          1.06        |
> > |               | ChatGLM4-Flash    | 60.48%           |           1.10    |
> > | DiffuScene| GPT-4o            | 68.37%           |       1.20        |
> > |               | Qwen-3-8B         | 61.92%           |        1.23          |
> > |               | ChatGLM4-Flash    | 59.20%           |          0.96        |
> >
> > From these results, we observe that scenes generated by OptiScene enable higher navigation success rates and lower navigation errors, despite all methods using the same downstream navigation policy in the instruction guide scenes. This provides stronger evidence of OptiScene’s superiority in generating layout structures that are more spatially coherent and traversable.
> >
> > In addition, as mentioned in our response to **R1Q3**, we also adopted a navigability metric to measure the structural connectivity of generated scenes (i.e., whether free space is connected and explorable). While all methods achieved high navigability, OptiScene still yielded more consistently connected spaces, further supporting its practical utility for embodied tasks.
> > In summary, our use of the downstream navigation task serves as a proxy to indirectly assess the functional quality and usability of the generated scenes. It is not intended as the main objective of our method, but rather as supporting evidence for layout quality.
> >
> > And at last, we agree that the applicability of OptiScene to more complex environments—such as multi-room layouts or scenes with corridors—is an important open direction. We are currently extending our framework to support such scenarios and will discuss this in the revision as part of future work.
> >
> > We sincerely thank the reviewer again for the constructive suggestions and thoughtful feedback. If there are any further questions or clarifications needed, we would be happy to continue the discussion.

---

> > ### Author Response · Authors · 2025-08-09
> > **Follow-up on our rebuttal responses for Reviewer LV9F**
> >
> > Dear Reviewer LV9F,
> >
> > We sincerely thank you for your professional and valuable reviews. Your insightful feedback will greatly help strengthen our paper.
> >
> > We have carefully addressed your latest questions. As the deadline is approaching, if you have any further concerns, please let us know, and we will do our best to respond promptly.
> >
> > Thank you again for your time and effort in reviewing our paper.

---

### Official Review · Reviewer_qyqm · 2025-07-03

**Clarity:** 3
**Significance:** 2
**Originality:** 3
**Rating:** 5
**Confidence:** 5

**Summary:**

The paper proposes to fix the gap between prompt driven LLM approaches which are costly and learning based methods which are constrained. First it proposes a new dataset-- 3D-SynthPlace to expand the 3D-Front dataset with synthetic samples generated with Holodeck and then filtered with a human in the loop system. The dataset not only doubles the size of the dataset but also increases the diversity of the scenes to include kitchens and bathrooms. The authors train an LLM on this dataset to output a structured JSON which captures the scene with scale and position of the objects. Rather than solely training the LLM using SFT, the authors choose to use RL based DPO fine-tuning for greater alignment with human preference by comparing against perturbed versions (collision) of the scene. The method is compared against 5 baselines and an ablation of the SFT vs RL is provided.

**Questions:**

* Does the structured JSON proposed here bring any advantage over other very similar representations proposed in related works?
* I'm very much on the borderline, please address weaknesses

**Ethical Concerns:**

["NO or VERY MINOR ethics concerns only"]

**Final Justification:**

The authors have done a good job with the rebuttal and have adressed my main concerns. I have recommened the authors to include some of the discussions/clarifications and evaluations to be included in the final revision. I recommend the work for acceptance and would like to see some small objects in the results.

**Limitations:**

Door and window positions are missing in the generated layouts. It should be discussed in limitations.

**Quality:**

2

**Strengths And Weaknesses:**

## Strengths
* The paper is well-written and all necessary details are provided.
* The results are quite strong, both qualitatively and quantitaively. The evaluation setup apart from LLM as a judge is robust and the ablation shows good gains in performance when tuning with RL. The additional user study paints a more complete picture.
* The method does not need to perform constrained optimization solve like FlairGPT or Holodeck which makes it computationally more feasible.
* Lower token cost compared to prompt driven methods & multi-agent frameworks
* The RL based fine-tuning with petrurbed scenes is ineteresting and lowers errors and object collisions (object overlap rate)

## Weaknesses
* Related works such as Holodeck and FlairGPT also take into account the door & window positions in the room and walls + objects on walls but this method misses out on this vital detail. Small objects make the scene much richer and provides a more complete layout of the room. Given that the authors use Holodeck for creating the synthetic dataset, I find the omission of small objects a limitation. The authors do not discuss this omission from Holodeck.
* It's well known in the community that SFT memorizes and RL generalizes [SFT Memorizes, RL Generalizes: A Comparative Study of Foundation Model Post-training, Chu et al], therefore the contribution is not particularly unique but it might be the first scene layout method to apply this (is this the case?). Nevertheless, I think the authors should mention this in their work since it's not a novel observation.
* The LLM as a judge (GPT-4o Ratings) for layout evaluation is not robust. It's well known that LLMs fail to provide reliable ratings in a 1-10 scale, compared to their text summaries which are considerably better.


## Neutral
* The dataset could have been a nice contribution given the human-in-the-loop system for filtering poor layouts. However, since the method disposes small objects, the dataset contribution becomes weaker.

---

> ### Author Rebuttal · Authors · 2025-07-31
>
> We sincerely thank you for the positive feedback. We appreciate your recognition of evaluation rigor, and the effectiveness of our RL-based fine-tuning.
> Below we address your questions.
>
> **R1Q1: Omitted small elements (doors, windows, wall‑mounted items) not addressed despite their inclusion in Holodeck/FlairGPT.**
>
> **Response**:
>
> First, our model is designed for indoor scene layout generation with a focus on placing large furniture items. Accordingly, we extended our dataset based on the 3D‑Front format. The original 3D‑Front does not include doors, windows, or wall‑mounted objects, so—for a fair comparison with prior methods—we omitted those elements during training. As in previous methods, DiffuScene, LLplace, etc., do not predict the wall-mounted objects, we follow this setting.
>
> However, when we generated synthetic scenes with Holodeck and applied post‑processing, we did retain the door/window and wall‑object annotations. In this setting, we then ran two new experiments: one trained solely on 3D‑SynthPlace data augmented with doors, windows, and other wall objects; and another trained on a mix of the original 3D‑Front data (without wall objects) and the 3D‑SynthPlace-with‑wall‑objects data. We then inference 100 times on four room categories. The results are demonstrated in Table 1：
>
> **Table 1: Effect of including small wall‑mounted elements on placement recall and overall layout quality.**
> |Training Data|avg. OOR Rate↓| avg. GPT4o ↑| UR ↑|
> |-|-|-|-|
> |3D‑Synth. (with wall objects)|0.037|8.7|63%|
> |Mixed 3D‑Front + 3D‑Synth. (with wall objects)|0.042|8.6|56%|
>
> While our method focuses on generating layouts with large furniture, we intentionally did not include small objects in our training data.
> Our primary focus is indoor scene layout generation, consistent with prior works such as **ATISS, DiffuScene, InstructScene, I-design, LLplace, LayoutVLM, and LayoutGPT**—all of which emphasize the placement of large furniture.
>  The arrangement of small objects (e.g., books, cups, desk clutter) is a fine-grained subtask that we consider **out of scope** for this work and plan to explore in future extensions.
> Our goal in constructing the dataset is to improve large-object layout modeling, which is foundational to scene generation. As shown in **Table 1 of our main paper**, our dataset significantly improves the performance of strong baselines such as LLplace and DiffuScene. This aligns with our core motivation: to augment and refine the 3D-FRONT dataset to support high-quality, human-aligned layout generation.
> While Holodeck includes many small items, their placements are often random and unrealistic, such as scattered desktop objects or items placed directly on the floor. Including such noisy placements would distract the model from learning coherent spatial structure.  Therefore, we intentionally excluded unfiltered small-object data from Holodeck to maintain semantic focus and layout consistency.
> But we think adding the windows/doors/wall-mounted objects can enhance scenes realistic, we will update the dataset with these and include Table 1 in the next version paper.
>
> **R1Q2. The observation that "SFT memorizes while RL generalizes" is not novel—does the contribution lie in applying it to scene layout generation for the first time?**
>
> **Response**:
>
> While it is well known that SFT tends to memorize its training data whereas reinforcement‑style methods generalize (Chu et al.), to our knowledge, OptiScene is the first work to integrate Direct Preference Optimization (DPO) on top of SFT specifically for 3D scene layout generation. We begin by fine‑tuning the model on the annotated dataset and then perform extensive human‑in‑the‑loop curation to extract only those layouts that reflect natural, human‑preferred living environments.
>
> In **Stage 1 DPO**, these curated positive examples teach the model to correct SFT’s memorization biases and prioritize realistic furniture arrangements—effectively aligning it with genuine human preferences. In **Stage 2 DPO**, we introduce controlled hard negatives— random translations, overlaps (e.g., a floor lamp inserted into a sofa; excluding overlaps common in normal scenes such as a chair under a desk), and minor out‑of‑bounds placements—to force the model to distinguish plausible layouts from artifacts. As reported in Table 6 of main paper, our two‑stage DPO pipeline delivers significant improvements in scene usability and substantially reduces collision and overlap errors compared to an SFT‑only baseline, demonstrating that DPO not only mitigates memorization but also produces layouts that more faithfully mirror real‑world living spaces.
>
> **R1Q3. Using LLMs (e.g., GPT-4o) as layout evaluators via 1–10 rating scales is unreliable—why do we use this, and what can we test more?**
>
> **Response**:
>
> Assessing the usability of generated scenes, which is closely aligned with human preference, is indeed a challenging task. In this field, leveraging GPT-based models for evaluation has become a widely adopted practice, nearly all recent works—including **LLplace, I-Design, and LayoutVLM**—have relied on GPT-4o for scoring. In our case, we do not simply feed the generated JSON layouts into GPT-4o. Instead, we render the scenes into images and then provide these images as input to GPT-4o for evaluation.
>
> We believe that thanks to GPT-4o’s extensive prior knowledge and alignment with human preferences, it provides a reasonable proxy for assessing output quality. However, we also acknowledge its limitations. The final score can be influenced by rendering quality, lighting, and camera perspectives, which may not directly reflect the underlying model’s generation capability.
>
> To address this, we complement GPT-4o scoring with additional metrics such as FID, OOR, and a human usability score to provide a more comprehensive evaluation of model performance. We agree on the need for a unified quality metric and are building a comprehensive benchmark, inspired by SceneEval, that uniformly renders each scene in Blender from a top‑down view and evaluates against the original JSON and user instruction. In addition to our existing OOR (object overlap rate), FID, GPT‑4o scores, and usability rate metrics, we will include:
>
> - CNT (Object Count): compare expected vs. detected object quantities by rendering each object’s front view and using Qwen2.5‑VL‑7B-Instruct to classify and count instances.
>
> - NAV (Navigability): compute the ratio of the largest connected free‑space region in a 2D occupancy grid.
>
> - OOB (Out‑of‑Bounds): analytic check of bounding boxes crossing room boundaries.
>
> - ACC (Accessibility): identify each object’s main functional face via Qwen2.5‑VL‑7B-Instruct and compute the unobstructed area in front of it.
>
> This streamlined suite combines rendering, JSON validation, and automated VLM/analytic checks to robustly assess scene completeness, correctness, and real‑world usability.  The model we tested is the version we retrained with the 3D-SynthPlace dataset:
>
> **Table 2:Comparison of DiffuScene, LLplace vs. our method across all benchmark metrics.**
> |Method| vg. OOR ↓|FID ↓|avg. GPT‑4o Score ↑| UR (%) ↑ | CNT Accuracy (%) ↑ | NAV (%) ↑ | OOB (%) ↓ | ACC (%) ↑ |
> |-|-|-|-|-|-|-|-|-|
> |**DiffuScene (w/ 3D-Synth)**|0.080|43.90|7.98|23.8|67.3|98.3|25.79|70.2|
> |**LLplace (w/ 3D-Synth)**|0.058|39.88|7.86|31.3| 97.8|98.8 |12.86|72.4|
> |**Ours**|0.025|33.68|8.90|50.0|98.6|99.6|8.58| 88.7|
>
> As shown in Table 2, our method outperforms DiffuScene and LLplace across all key metrics, achieving lower overlap and FID, higher GPT‑4o scores and usability, as well as stronger CNT, NAV, OOB, and ACC performance. This demonstrates the effectiveness and robustness of our proposed approach. We will include all of these metrics in the appendix of the future version. However, scene generation research still lacks a clear, unified benchmark, so we are developing a new evaluation suite that builds on our current metrics by adding curated test sets with standardized rendering protocols, rigorous FID comparisons under consistent conditions, and a CLIP‑based evaluator fine‑tuned on paired scene images and text instructions—together providing a more robust and reproducible standard for future work.
>
> **R1Q4: Does the structured JSON proposed here bring any advantage over other very similar representations proposed in related works?**
>
> **Response**:
>
> Thank you for the question. We chose a structured JSON format for representing 3D scene layouts because it offers an **effective balance between human readability and machine interpretability**. The format uses clear key-value pairs such as "x": 1.2 and "rotation": 90, which are easy to read and understand. At the same time, JSON supports schema validation, type checking, and straightforward extensibility, allowing new fields to be added without disrupting existing workflows.
>
> Unlike formats such as GLB or USD, which tightly integrate geometry, textures, and scene graphs, JSON separates spatial properties from rendering details. This separation makes it particularly suitable for large-scale layout generation and editing, where assets can be dynamically loaded and replaced in engines like Isaac Sim or Blender.
> In contrast to CSS-style in LayoutGPT, 2D layout representations that rely on properties such as left, top, width, and height and often require complex transformation logic to describe spatial arrangements, JSON provides a more precise and semantically rich encoding of full 3D information, including position, rotation, and size. It is also inherently more extensible, allowing new spatial attributes or metadata to be added without altering the existing structure.
>
> Most importantly, JSON serves as a practical intermediate format between natural language and executable code. It enables LLM to convert user instructions into structured scene layouts that can be directly used in downstream tasks such as 3D modelling and rendering in Blender or Isaac Sim.

---

> ### Author Response · Authors · 2025-08-05
>
> Dear Reviewer,
>
> I hope this message finds you well. As the discussion period is nearing its end, I wanted to ensure we have addressed all your concerns satisfactorily. If there are any additional points or feedback you'd like us to consider, please let us know. Your insights are invaluable to us, and we’re eager to address any remaining issues to improve our work.
>
> Thank you for your time and effort in reviewing our paper.

---

> ### Comment · Reviewer_qyqm · 2025-08-05
>
> Dear authors,
>
> Thanks for the rebuttal! I am satisfied with the resonse and shall be increasing my score. I would recommend including these discussions and evalaution in the paper to strengthen it.
> In the revision, please cite past works which have studied "SFT memorizes while RL generalizes" and explictly state the JSON format is used in other works and have inspired the method.  I do not think including them would diminish the novely aspects, as the authors correcly state that it has not been studied in this particular context for RL. I would also advise adding some results with small objects.

---

> > ### Author Response · Authors · 2025-08-06
> >
> > We sincerely thank the reviewer for the positive feedback and for the decision to raise the score. We greatly appreciate your thoughtful suggestions for improving the paper.
> >
> > In the revised version, we will incorporate the recommended discussions and evaluations. Specifically, we will (1) cite relevant prior works that have studied the distinction between “SFT memorizes while RL generalizes”; (2) clearly state that the use of a structured JSON format is inspired by existing methods in the literature, while emphasizing that our formulation and application in the RL context remain novel;(3) include additional evaluation results involving door, windows and wall-mounted objects to further strengthen the empirical analysis and we will include more small objects in the future work!
> >
> > We are grateful for your constructive comments and will ensure that these points are carefully addressed in the final version.

---

### Note · Authors · 2025-08-11

We sincerely thank the Reviewers, AC, and SAC for the time and effort.

After the rebuttal, two reviewers explicitly indicated they raised scores, and all assessments are positive. All reviewers maintained a high level of professionalism and confidence throughout the process.
Before the rebuttal, reviewers had already recognized key strengths of our work on indoor 3D scene layout generation: the scale and quality of 3D-SynthPlace, effective two-stage SFT+DPO alignment, robust experiments, lower compute/token cost with fewer collisions/out-of-room errors than constraint- or agent-heavy baselines, and clear potential for downstream tasks.
We are grateful for the reviewers' recognition, and here are the details of each reviewer:

- Reviewer qyqm. All concerns were addressed. The reviewer confirmed a score increase and affirmed our novelty. We clarified our first use of RL for human-preference alignment in scene generation, situated the structured-JSON design relative to prior work, and added evaluations on doors, windows, and wall-mounted objects.
- Reviewer LV9M. Starting from a score of 4, the reviewer emphasized our clear contribution after the rebuttal, noting that results significantly outperform baselines. Although they questioned the navigation evaluation, we clarified that it serves to demonstrate the usability of generated layouts (not the main task) and provided additional comparisons for the AC’s and reviewer’s reference.
- Reviewer xtFJ. Our responses resolved key doubts and led to a higher score. We added implementation details (SFT supervision, the wrapper) and explained how two-stage DPO handles legal bounding-box overlaps. We also included instruction-following tests demonstrating reliability and generalization. These will be included in the future paper version.
- Reviewer 1Sit. With an initially positive score, the reviewer strongly endorsed the rebuttal and noted that most concerns were addressed. At the reviewer’s request, we further evaluated OOD objects, explicit spatial relations, and room–object mismatched user instructions, and we obtained positive results.

In summary, all reviewer assessments are positive, and the overall feedback is strongly supportive. We believe we have addressed nearly all concerns. For the remaining points, please refer to our follow-up discussions, and we hope the additional experiments address any remaining concerns. We hope our work and rebuttal meet the reviewers’ and AC’s expectations.

---

### Decision · Program_Chairs · 2025-09-17

**Decision:**

Accept (poster)

**Comment:**

The paper proposes OptiScene, a new system for indoor scene layout generation from natural-language description. OptiScene is an LLM-based framework that is fine-tuned through supervised fine-tuning (SFT) and RL (specifically, 2-stage Direct Preference Optimization [DPO]). It is trained on their 17,000 scene dataset called 3D-SynthPlace that is based on the 3D-Front dataset and Holodeck-generated synthetic layouts. OptiScene also has the potential to be used for scene editing and object-centric navigation.

Pros: OptiScene’s novel approach combines the strengths of prompt-driven and learning-based methods, and aligns the generation process with human preferences. The paper also introduces a new and useful dataset 3D-SynthPlace. SOTA results validate the design decisions.

Cons: Though the architecture in combination with its application space is new, the basic ideas of SFT memorizes and RL generalizes are not new. Also, using LLMs (e.g., GPT-4o) as layout evaluators via 1–10 rating scales tends to be unreliable. (This problem may be ameliorated by using additional metrics such as FID and OOR.)

There is consensus among the reviewers to accept. They like the technical contributions of OptiScene and 3D-SynthPlace, and SOTA results shown support their novel system.

The major issues raised by reviewers were addressed by the authors. Some examples: possible bias caused by using Holodeck, performance with varying scene complexity based on object count, no ablation study on GPT4o, FID, and user instruction complexity. Responses: bias is minimized via manual filtering and refinements, and results on scene complexity and the suggested additional ablation studies were given. On the novelty front, the authors reiterated that OptiScene is the first to integrate DPO on top of SFT for 3D scene layout generation. This is in response to a comment that it is well-known that SFT tends to memorize its training data while reinforcement-style methods generalize.